# Arboviruses and symbiotic viruses cooperatively hijack insect sperm-specific proteins for paternal transmission

Jiajia Wan[1,3], Qifu Liang[1,3], Ruonan Zhang[1,3], Yu Cheng[1], Xin Wang[1], Hui Wang[1], Jieting Zhang[1], Dongsheng Jia[1], Yu Du[1], Wenhui Zheng[1], Dingzhong Tang [2], Taiyun Wei [1] ✉ & Qian Chen [1] ✉

Arboviruses and symbiotic viruses can be paternally transmitted by male insects to their offspring for long-term viral persistence in nature, but the mechanism remains largely unknown. Here, we identify the sperm-specific serpin protein HongrES1 of leafhopper *Recilia dorsalis* as a mediator of paternal transmission of the reovirus Rice gall dwarf virus (RGDV) and a previously undescribed symbiotic virus of the *Virgaviridae* family, Recilia dorsalis filamentous virus (RdFV). We show that HongrES1 mediates the direct binding of virions to leafhopper sperm surfaces and subsequent paternal transmission via interaction with both viral capsid proteins. Direct interaction of viral capsid proteins mediates simultaneously invasion of two viruses into male reproductive organs. Moreover, arbovirus activates HongrES1 expression to suppress the conversion of prophenoloxidase to active phenoloxidase, potentially producing a mild antiviral melanization defense. Paternal virus transmission scarcely affects offspring fitness. These findings provide insights into how different viruses cooperatively hijack insect sperm-specific proteins for paternal transmission without disturbing sperm functions.

Insects are the host and vector of diverse viruses including those that infect vertebrates, plants, and fungi. Insect symbiotic viruses reside inside their insect hosts and are passed vertically from parent to offspring, suggesting that the viruses could invade host germ cells[1–4]. Insect symbiotic viruses are generally considered to be maternally or paternally transmitted to host offspring[1,3,4]. For example, Sigma virus in dipterous insects, cell fusing agent virus and Galbut virus in *Aedes aegypti*, Gypsy retrovirus in *Drosophila*, and Diaphorina citri reovirus (DcRV) in psyllid host can be paternally transmitted to offspring[2,5,6]. Pteromalus puparum negative-strand RNA virus 1 in its parasitoid wasp host can even infect the male reproductive system and transmit to offspring via mating, finally regulating the offspring sex ratio[7]. Vertical transmission of insect symbiotic viruses is an important persistence model in nature[1,8].

Currently, whether or how insect symbiotic viruses exploit sperms for paternal transmission remains poorly understood.

Arthropod-borne viruses (arboviruses) that cause substantial global health or agricultural problems can also be paternally transmitted to insect vector progeny population[5,9–11]. For example, La Crosse virus and Zika virus can be paternally transmitted by male mosquitoes[12–15]. Maternal arbovirus transmission through transovarial passage has been extensively investigated[16–21], but how sperm-mediated paternal virus transmission occurs is not completely clear. Considering the sperm structure, arbovirus infection is expected to impair normal functions of mammal host sperm[22,23]. We have revealed that rice gall dwarf virus (RGDV), a plant reovirus, can be paternally transmitted from male leafhopper vectors to offspring through a direct association of viral outer protein with the sperm surface without

[1]Vector-Borne Virus Research Center, Fujian Agriculture and Forestry University, Fuzhou, Fujian 350002, China. [2]State Key Laboratory of Ecological Pest Control for Fujian and Taiwan Crops, Fujian Agriculture and Forestry University, Fuzhou, Fujian 350002, China. [3]These authors contributed equally: Jiajia Wan, Qifu Liang, Ruonan Zhang. ✉e-mail: weitaiyun@fafu.edu.cn; chenqian@fafu.edu.cn

disturbing sperm functions[24]. During maternal transmission, RGDV propagation in the oocytes of the female ovary often causes cyto-pathologic changes to decrease insect offspring fitness[24,25]. By contrast, sperm-mediated paternal transmission of RGDV scarcely affects insect offspring fitness[24]. Thus, paternal arbovirus transmission may have evolved as a preferred mode of vertical transmission during the long-term virus-vector interaction.

We previously revealed that major outer capsid protein P8 of RGDV interacts with heparan sulfate proteoglycan (HSPG) to facilitate sperm-mediated paternal virus transmission[24]. However, HSPG is composed of unbranched, negatively charged heparan sulfate polysaccharides attached to a variety of cell surface or extracellular matrix proteins[26]. Insect sperm- or semen-specific components potentially mediate paternal virus transmission, but the mechanism remains unrevealed. Many sperm surface proteins have critical roles in various reproductive processes such as sperm maturation, capacitation, and sperm-egg recognition[27]. In mammals, the epididymis is responsible for sperm transport, concentration, storage, and maturation[28]. HongrES1 and Spink13, the members of the serpin (serine proteinase inhibitor) family, are specifically expressed in the cauda epididymidis, deposited on the sperm surface, and involved in sperm capacitation and maturation[29–31]. Thus, serpin proteins appear to be necessary for the regulation of critical serine protease activity during sperm fertilization and maturation. In insects, serpin proteins are also specifically present in the seminal fluid of mating males, but their functions remain undescribed[32–34]. Insect serpin proteins are involved in antiviral melanization process by negatively regulating the clip-domain serine proteases activity, which mediating the conversion of the zymogen prophenoloxidase (PPO) into active phenoloxidase (PO)[35]. Subsequently, PO mediates the formation of melanin, which directly encapsulates and kills certain pathogens[36,37]. In this study, we report that RGDV and a symbiotic virus in rice green leafhopper *Recilia dorsalis* activate and hijack HongrES1 homolog on sperm surface for paternal virus transmission without disturbing sperm function. Furthermore, the elevation of HongrES1 expression during viral infection effectively suppresses the production of excessive PO contents to benefit viral infection in insect male reproductive system. Our findings reveal a new mode for arboviruses and symbiotic viruses to cooperatively hijack insect sperm-specific proteins for co-paternal transmission.

## Results

### The vertical transmission of a symbiotic virus of *R. dorsalis*

We identified a previously undescribed Recilia dorsalis filamentous virus (RdFV) belonging to the *Virgaviridae* family from field-caught *R. dorsalis* population in Guangdong Province, Southern China in 2018, based on analysis of RNA sequencing data (Fig. S1). The (+) single-stranded RNA genome of RdFV was 16,158 nt in length (GenBank accession No. OP326514) and contained 6 open reading frames (ORFs) to encode 6 proteins, including RNA-dependent RNA polymerase (RdRp) and capsid protein (CP) (Fig. 1a). Phylogenetic analysis based on amino acid sequences of RdRp showed that RdFV clustered with plant-infecting furoviruses in *Virgaviridae* family, which contained plant virgavirus lineages and unclassified insect-infecting virga-like virus group, such as Hubei virga-like viruses[38–40] (Fig. S1). RdFV possessed a high incidence of about 80% in lab-reared male or female *R. dorsalis* population (Fig. 1b, c). Immunofluorescence microscopy assays showed that RdFV CP was distributed in the midgut or salivary glands of virus-positive *R. dorsalis* (Fig. S2). Immunoelectron microscopy confirmed that RdFV CP antibody specifically reacted with the filamentous viral particles in the cytoplasm of the midgut epithelium (Fig. 1d, e).

To explore whether RdFV can be vertically transmitted, the offspring of four mating pairings, including infected unfemales × uninfected males; infected females × uninfected males; uninfected females × infected males; and infected female × infected males, were collected and analyzed. We observed about 52%, 67%, and 85% positive for RdFV from the offspring of infected females mating with uninfected males, uninfected females mating with infected males, and infected females mating with infected males, respectively (Fig. 1f). Thus, paternal transmission of RdFV was more efficient than maternal transmission. Immunofluorescence microscopy showed that RdFV CP distributed in male testes and female ovaries (Figs. S2 and S3). Furthermore, RdFV CP was closely associated with most of the sperms directly dissected from infected male testes (Fig. 1g, h). Immunoelectron microscopy confirmed that RdFV CP antibody specifically reacted with the filamentous viral particles in the germ cells of female ovary (Fig. S3), or associated with the sperms in male testes (Fig. 1i, j). Such association of RdFV particles with sperms did not affect sperm morphology (Fig. S4). Furthermore, immunoelectron microscopy showed that virus-associated sperms were present in the dissected spermatheca of uninfected females at 5-day post mating with infected males (Fig. S5), confirming the transfer of virus-associated sperms from infected males to females. Mating experiments shown above (Fig. 1f) suggested that such RdFV-associated sperms in the spermatheca finally fertilized the mature eggs during ovulation. We thus determine that RdFV can be vertically transmitted in *R. dorsalis* via transovarial or sperm-mediated paternal transmission.

To investigate the effect of RdFV infection on the fitness of *R. dorsalis* population, the eggs produced by two mating pairings, including one RdFV-positive virgin female × one RdFV-positive male and one RdFV-free virgin female × one RdFV-free male, were collected and analyzed. It was found that RdFV prolonged adult longevity and promoted egg development (Fig. S6). For example, the mean longevity of the RdFV-positive male population (22 d) was about 4 d longer than that of the RdFV-free male population (18 d) (Fig. S6a). Moreover, RdFV-positive parents laid about 20% more eggs compared to RdFV-free parents (Fig. S6b). More importantly, the RdFV-positive eggs developed better than RdFV-free controls (Fig. S6c). The mean development duration for virus-positive eggs was reduced by about 2 days (Fig. S6d), and the hatching rate for RdFV-positive eggs was increased by about 10% at 11 days after oviposition (Fig. S6e). Thus, RdFV infection benefits the fitness of host adults and their offspring, and thus does not affect the functions of host sperm or ovary.

### RdFV or RGDV activates HongrES1 in male reproductive system of *R. dorsalis*

We then used RdFV CP as a bait protein to screen its interactors of *R. dorsalis* from a yeast cDNA library of *R. dorsalis* by using yeast two-hybrid system (Y2H). Sequencing of the positive clones identified HongrES1 homolog of *R. dorsalis*. The full-length ORF of *HongrES1* contained 1,590 nt and encoded 415 amino acid residues including a serpin (serine protease inhibitor) conserved domain (Fig. S7). Y2H and glutathione *S*-transferase (GST) pull-down assays confirmed that RdFV CP interacted with HongrES1 (Fig. 2a, b). Previously, we showed that RGDV P8 was also closely associated with the sperm head of *R. dorsalis*[24]. Similarly, Y2H and GST pull-down assays demonstrated the specific interaction of RGDV P8 with HongrES1 (Fig. 2a, b). A tissue-specific analysis indicated that HongrES1 was specifically expressed in the male reproductive system but not in other organs (Fig. 2c, d). Immunoelectron microscopy showed that HongrES1 antibody specifically reacted with the sperm surface in the male testis (Fig. 2e). RT-qPCR and western blot assays showed that RdFV infection significantly increased HongrES1 expression in the male reproductive system (Fig. 2f, g). Because RdFV possessed a high incidence in *R. dorsalis* population (Fig. 1b), we used RdFV-positive *R. dorsalis* population to investigate the relationship between RGDV and HongrES1. RT-qPCR and western blot assays showed that RGDV infection also significantly increased HongrES1 expression in the male reproductive system

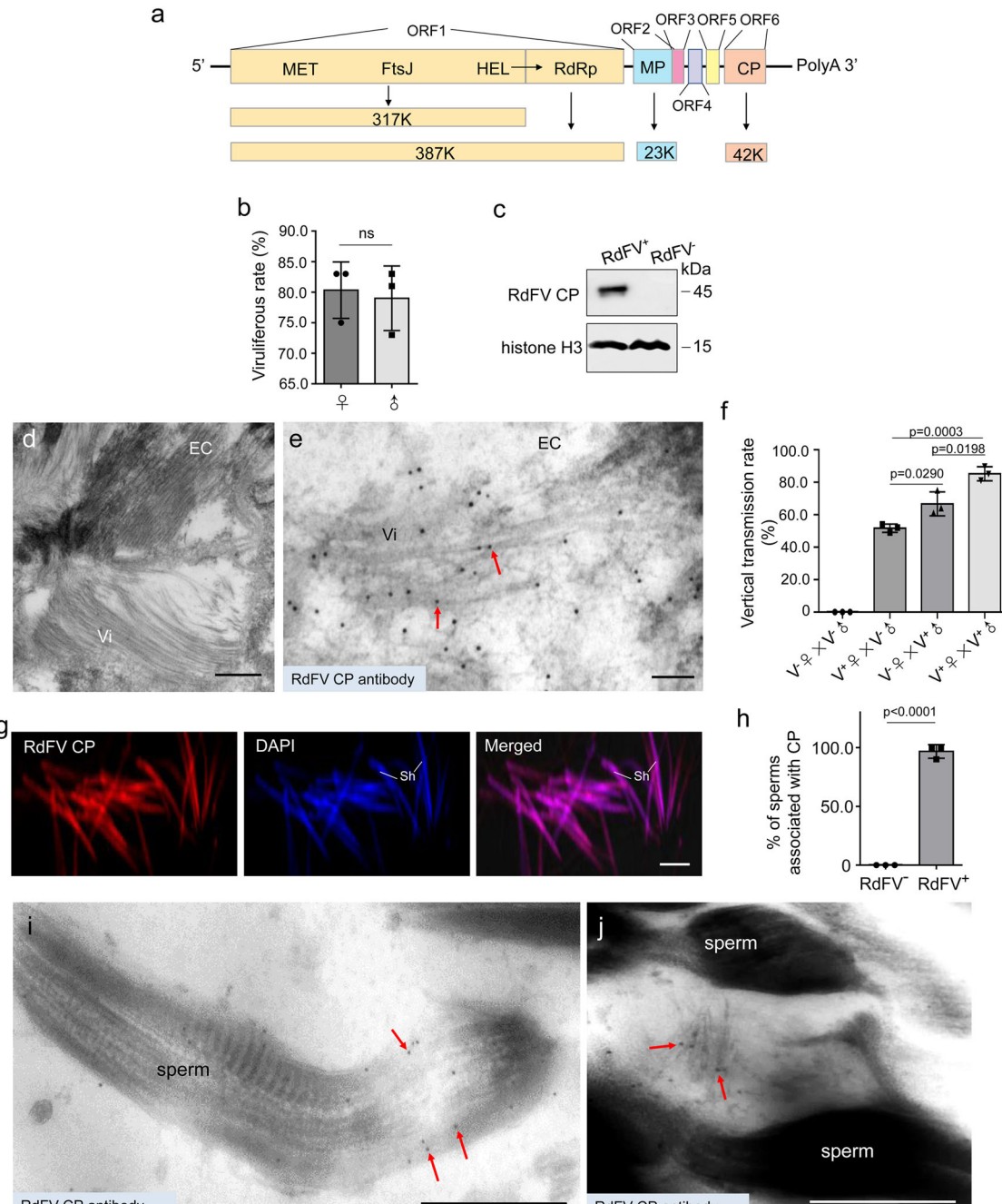

**Fig. 1 | Sperm-mediated paternal transmission of RdFV by male *R. dorsalis*.**
**a** Diagram of the genome organization of RdFV. Solid lines represent RNA; boxes indicate predicted ORFs. MET, methyltransferase; FtsJ, ftsJ-like methylthransferase; HEL, superfamily 1 helicase; RdRp, RNA-dependent RNA polymerase; MP, movement protein; CP, capsid protein. **b** Viruliferous rates in 100 male or female adults by RT-PCR assays for testing the transcript of RdFV *CP*. Means (± SD) represent three biological replicates (two-tailed t-test). Ns, not significant. **c** Accumulation of RdFV CP in 30 RdFV-positive or free leafhoppers, as determined by western blot assays by using HongrES1 or histone H3 antibody. Data represent three biological replicates. **d** Electron micrographs showing filamentous viral particles within the midgut epithelium of RdFV-positive leafhoppers. Bar, 500 nm. **e** Immunoelectron micrographs showing the localization of RdFV CP on filamentous viral particles within the midgut epithelium of RdFV-positive leafhoppers. EC, epithelial cell. Vi, virions. Bar, 100 nm. **f** Vertical transmission rates of RdFV caused by RdFV-positive

and free female or male leafhoppers via mating. The offspring were tested for presence of RdFV. V⁻, RdFV-free. V⁺, RdFV-positive. Fifty pairs were performed for each mating combination. Means (±SD) are shown from 50 offspring of each mating combinations and represent three biological replicates (two-tailed t-test). **g** RdFV CP distribution on the sperm head in vivo, as determined by immunofluorescence microscopy. The dissected sperms of RdFV-positive males were stained with RdFV CP-rhodamine (red) and nuclear dye DAPI (blue) to indicate sperm heads. Sh, sperm head. Bar, 10 μm. **h** The mean percentages of sperms immunolabeled by CP antibody (n = 100). Means (±SD) represent three biological replicates (two-tailed t-test). **i, j** Immunoelectron micrographs showing the localization of RdFV CP on filamentous viral particles in RdFV-positive testes. Bars, 500 nm. The intestines (**e**) or testes (**i, j**) of leafhoppers were immunolabeled with RdFV CP-specific IgG as the primary antibody, followed by treatment with 15-nm gold particle-conjugated IgG as the secondary antibody. Red arrows indicate gold particles.

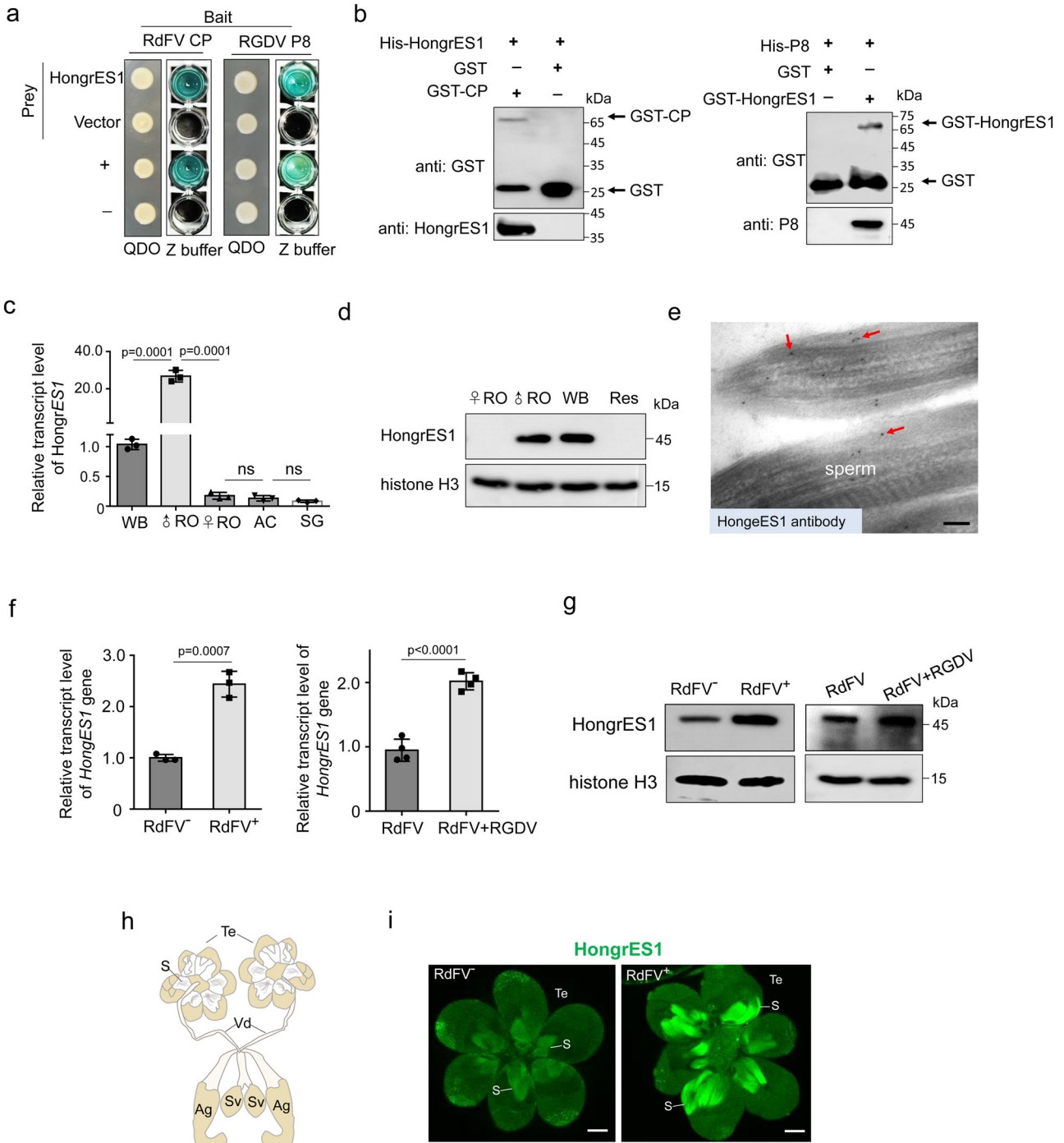

**Fig. 2 | Sperm-specific HongrES1 of *R. dorsalis* interacts with RdFV CP or RGDV P8. a** Y2H assays showing interaction of HongrES1 with RdFV CP or RGDV P8. Transformants on QDO plates are labeled as follows: HongrES1, pPR3-N-HongrES1/pDHB1-CP or -P8; vector, pPR3-N / pDHB1-CP or -P8; +, positive control, i.e., pLargeT/p53; −, negative control, i.e., pDHB1 / pRR3N. β-galactosidase assay was detected in Z buffer with X-Gal. QDO, SD/-Trp-Leu-His-Ade medium. **b** GST pull-down assays showing interaction of HongrES1 with RdFV CP or RGDV P8. **c**, **d** HongrES1 expression levels in different organs of 30 RdFV-positive leafhoppers, as determined by RT-qPCR (**c**) and western blot (**d**) assays, respectively. Means (± SD) are shown from three biological replicates (two-tailed t-test). Ns, not significant. WB, whole bodies. ♂RO, male reproductive organs. ♀RO, female reproductive organs. AC, alimentary canals. SG, salivary glands. Res, residues. **e** Localization of HongrES1 on sperm surface in male testis, as determined by immunoelectron microscopy. Testes of leafhoppers were immunolabeled with

HongrES1-specific IgG as the primary antibody, followed by treatment with 15-nm gold particle-conjugated IgG as the secondary antibody. Red arrows indicate gold particles. Bar, 100 nm. Effect of RdFV or RGDV infection on the expression of HongrES1 in male reproductive organs from 30 RdFV-free, RdFV-positive, or RdFV and RGDV co-positive leafhoppers, as determined by RT-qPCR (**f**) and western blot (**g**) assays. Means (±SD) are shown from at least three biological replicates (two-tailed t-test). The proteins were detected by using HongrES1 or histone H3 antibody in western blot assays. Data in **d** and **g** represent three biological replicates. **h** Schematic illustration of male reproductive system of *R. dorsalis*. Ag, accessory gland. S, sperm. Sv, seminal vesicle. Te, testis. Vd, vas deferens. **i** Distribution of HongrES1 in RdFV-free or positive testes, as determined by immunofluorecense microscopy. The testes dissected from 30 RdFV-free or positive males were immunostained with HongrES1-FITC (green). S, sperm. Te, testis. Bars, 10 μm.

(Fig. 2f, g). Immunofluorescence microscopy showed that RdFV infection triggered the increased accumulation of HongrES1, which was visible as the bundle of filamentous structures in all infected testes examined (Fig. 2h, i). Potentially, RdFV and RGDV could activate and hijack HongrES1 for their co-paternal transmission.

## Direct interaction of HongrES1 with RdFV CP or RGDV P8 mediates viral binding to sperms

We then investigated how RdFV or RGDV hijacked HongrES1 to attach sperms. Immunofluorescence microscopy showed that RdFV CP formed obvious punctate and filamentous structures, which could extensively distribute on HongrES1-decorated sperms in infected testes (Fig. 3a). Similarly, RGDV P8 formed obvious punctate structures to extensively distribute on HongrES1-decorated sperms in infected testes (Fig. 3b). We calculated that more than a half of CP- or P8-formed structures were colocalized with HongrES1 on sperms in infected testes (Fig. 3c). We then directly dissected sperms from RdFV-infected testes and observed that most of sperms were associated with HongrES1 and CP (Fig. 3d–f). Similarly, most of sperms directly dissected from RGDV-infected testes were also associated with HongrES1 and P8 (Fig. 3e–f). Collectively, these results indicate that the specific binding of RdFV or RGDV virions to the sperms is mediated by CP-HongrES1 or P8-HongrES1 interaction.

We further investigated how HongrES1 mediated the association of RGDV virions with the plasma membrane of sperms. Electron microscopy showed the association of RGDV icosahedral virions or virus-containing tubules with the plasma membrane of sperms in RGDV-infected testes (Fig. 3g-i). HongrES1 antibody specifically reacted with RGDV particles which distributed close to the sperms (Fig. 3j). More importantly, HongrES1 was densely distributed on sperm surfaces where RGDV virions or RGDV-induced tubules were attached (Fig. 3k-m). Such association of RGDV particles with sperms did not affect sperm morphology (Fig. 3g-m). It was clear that HongrES1 on the sperm surface served as the docking sites for RGDV particles attachment. Previously, we showed that RGDV virions-associated sperms distributed in the spermatheca of uninfected females after mating with infected males[24]. Thus, our electron microscopic observations further confirm that HongrES1-P8 interaction directly mediates the binding of RGDV particles with the sperm surface.

## HongrES1 promotes sperm-mediated RdFV or RGDV paternal transmission

We then investigated how RdFV or RGDV hijacked HongrES1 for paternal transmission. In the neutralizing virus-sperm binding experiment, in vitro binding of RdFV CP to live sperms was detectable after 1-h incubation, and sperm binding to RdFV CP was reduced by about 55% after pretreatment with HongrES1 antibody (Fig. 4a, b). Similarly, we detected the binding of purified RGDV virions to the sperms after 1-h incubation, and RGDV-sperm association was reduced by about 56% after pretreatment with HongrES1 antibody (Fig. 4c, d). It is indicated that HongrES1 mediates the direct binding of RdFV or RGDV particles to the sperms.

We then knocked down *HongrES1* expression by microinjecting males with synthesized dsRNAs targeting *HongrES1* (dsHongrES1) to investigate how HongrES1 promoted paternal virus transmission. RT-qPCR and western blot assays showed that the knockdown of *HongrES1* expression effectively decreased RdFV CP or RGDV P8 accumulation in the male reproductive system (Fig. 4e–h). Moreover, the rates of paternal RdFV or RGDV transmission were decreased by about 30% or 35% after dsHongrES1 treatment (Fig. 4i, j). It is clear that RdFV or RGDV can trigger the increased accumulation of HongrES1 in male reproductive system to benefit paternal virus transmission. Thus, the exploitation of HongrES1 is a conserved mechanism for the efficient parental transmission of RdFV and RGDV in *R. dorsalis* population.

## Direct interaction of viral capsid proteins mediates simultaneous invasion of two viruses into male reproductive organs and subsequent co-paternal transmission

We further investigated whether RdFV affected RGDV infection in male reproductive system of *R. dorsalis*. RT-qPCR assays showed that the transcript levels of RGDV *P8* in RdFV and RGDV co-positive male population were significantly higher than that in RdFV-free/RGDV-positive control (Fig. 5a). Mating experiments showed that paternal transmission of RGDV was more efficient in RdFV-positive male population (~80%) than in RdFV-free male control (~62%) (Fig. 5b). Thus, RdFV is beneficial for RGDV propagation and transmission in *R. dorsalis*.

In RdFV and RGDV co-positive testes, immunoelectron microscopy showed that the spherical particles of RGDV and the filamentous particles of RdFV were accumulated in the cytoplasm of testis epithelium, and HongrES1 antibody could specifically react with both viral particles (Fig. 5c, d). The colocalization of RGDV and RdFV particles in testis epithelium to form the complex was frequently observed (Fig. 5e, f). It appeared that the RGDV-RdFV complex was released into sperm-accumulated lumen via the apical plasmalemma (Fig. 5g, h), and then was clearly in contact with sperm surfaces (Fig. 5i). In RdFV and RGDV co-positive testes, immunofluorescence microscopy showed that about half of RGDV P8-formed puncta were co-localized with the puncta of RdFV CP or directly bound to the edges of filamentous structures of CP throughout virus-infected testes (Fig. 5j–l). Rice stripe mosaic virus (RSMV), a plant cytorhabdovirus, and rice dwarf virus (RDV), also a plant reovirus, are transmitted by *R. dorsalis* in a propagative-persistent manner, but not via paternal transmission manner[16,41]. Y2H assay indicated that RdFV CP specifically interacted with RGDV P8, rather than with RSMV glycoprotein (G) and RDV P8 (Figs. 5m and S8). GST pull-down assay confirmed the specific interaction of RdFV CP with RGDV P8 (Fig. 5n). Thus, RdFV virions could specifically bind to RGDV virions during the invasion of both viruses from the testis epithelium into sperm-accumulated lumen to simultaneously attach sperm surfaces in male testes of *R. dorsalis*.

In RdFV and RGDV co-positive male *R. dorsalis*, the knockdown of RdFV CP expression by microinjecting synthesized dsRNAs targeting *CP* (dsCP) significantly reduced RGDV and HongrES1 accumulation in the male reproductive system, finally decreasing paternal RGDV transmission rate by about 25% (Fig. 6a–f). Similarly, the knockdown of RGDV *P8* expression by microinjecting synthesized dsRNAs targeting *P8* (dsP8) also significantly reduced RdFV and HongrES1 accumulation in the reproductive system of RdFV and RGDV co-positive male *R. dorsalis*, and decreased paternal RdFV transmission rate by about 22% (Fig. 6a–f). Because RdFV or RGDV infection activates HongrES1 expression, and thus the knockdown of RGDV P8 or RdFV CP expression accordingly decreases HongeES1 accumulation, which finally inhibits viral infection in male reproductive system and subsequent paternal virus transmission. Taken together, our results suggest that the direct interaction of viral capsid proteins mediates simultaneous invasion of two viruses into male reproductive organs to attach sperms, and RdFV and RGDV could collaboratively activate HongrES1 for co-paternal transmission (Fig. 6g).

## HongrES1 effectively suppresses the production of excessive PO contents to benefit viral infection in insect male reproductive system

We further investigated the function of HongrES1 during RGDV infection in male reproductive system of leafhoppers. Insect antiviral melanization defense consists of a cascade of clip-domain serine proteases that converts the zymogen PPO to active PO, which is negatively regulated by serpins[42,43]. RT-qPCR assays showed that the transcript levels of most *clip-domain serine protease* genes were decreased in RGDV-positive male *R. dorsalis* (Fig. 7a). We then investigated whether HongrES1 could modulate the PO activity in male reproductive system

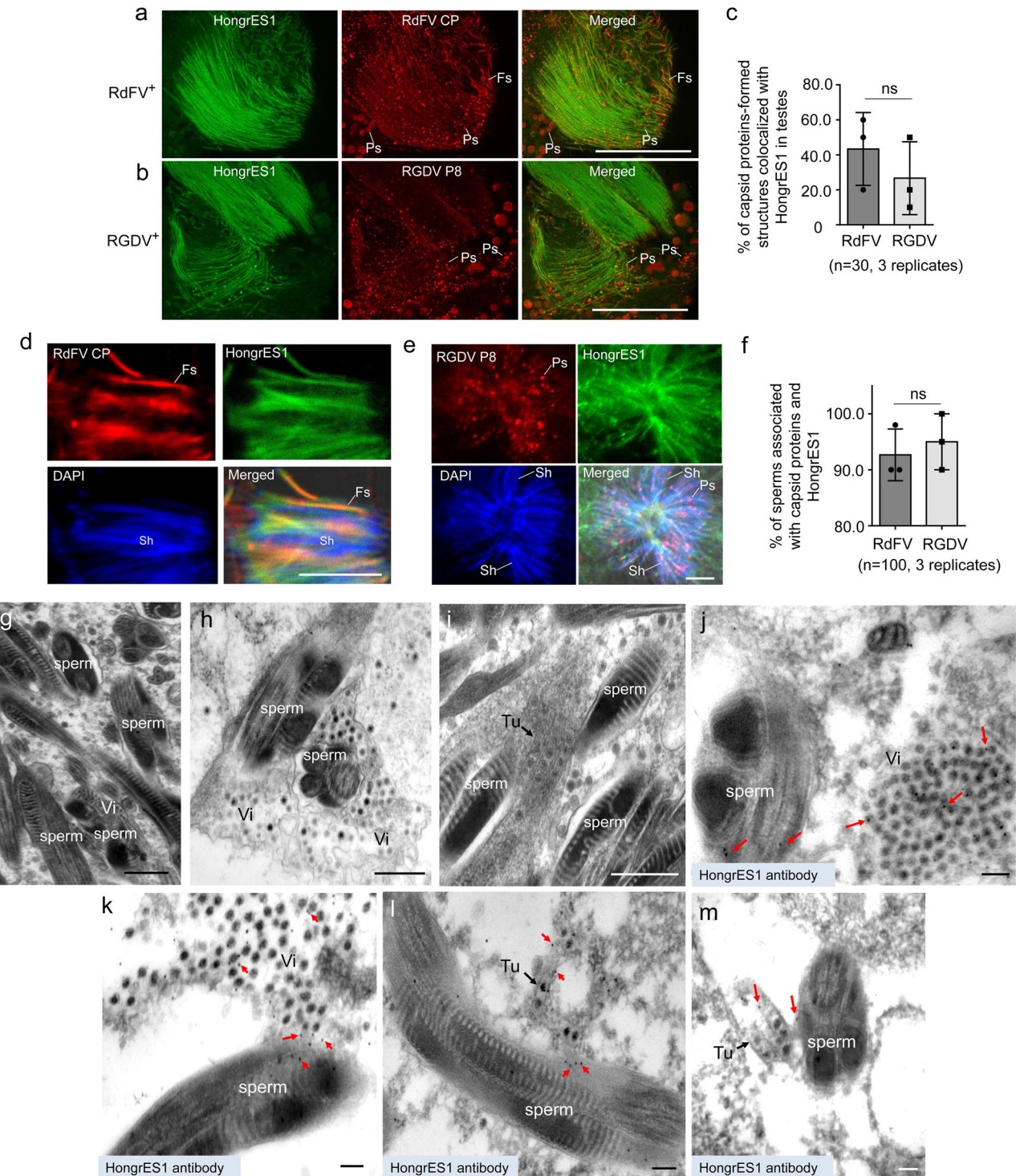

**Fig. 3 | Direct association of RdFV CP or RGDV P8 with HongrES1 on sperms in infected testes of male _R. dorsalis_.** Association of RdFV CP (**a**) or RGDV P8 (**b**) with HongrES1, as determined by immunofluorecense microscopy. The dissected testes from 30 RdFV- or RGDV-positive males were immunostained with HongrES1-FITC (green) and CP-rhodamine (red) (**a**), or with HongrES1-FITC (green) and P8-rhodamine (red) (**b**). Ps, punctate structure. Fs, filamentous structure. Bars, 10 μm. **c** The mean percentages of CP- or P8-formed structures colocalized with HongrES1 in respective infected testes (_n_ = 30). Means (±SD) represent three biological replicates (two-tailed t-test). Ns, not significant. Association of RdFV CP or RGDV P8 with HongeES1 on the dissected sperms, as determined by immunofluorecense microscopy. The dissected sperms from 30 RdFV- or RGDV-positive testes were stained with HongrES1-FITC (green), CP-rhodamine (red) and nuclear dye DAPI

(blue) to indicate sperm head (**d**), or with HongrES1-FITC (green), P8-rhodamine (red) and DAPI (blue) (**e**). Sh, sperm head. Bars, 10 μm. **f** The mean percentages of sperms (_n_ = 100) associated with capsid proteins and HongrES1. Means (±SD) represent three biological replicates (two-tailed t-test). Ns, not significant. **g** Electron micrograph showing the sperms in virus-free testes. Bar, 500 nm. Electron micrograph showing the association of RGDV virions (**h**) or virus-containing tubules (**i**) with the plasma membrane of sperms in RGDV-infected testes. Bars, 500 nm. **j**–**m** Immunoelectron microscopy showing the association of HongrES1 with RGDV particles and sperms. Male testes were immunolabeled with HongrES1-specific IgG as the primary antibody, followed by treatment with 15-nm gold particle-conjugated IgG as the secondary antibody. Red arrows indicate gold particles. Vi, virions; Tu, tubule. Bars, 100 nm.

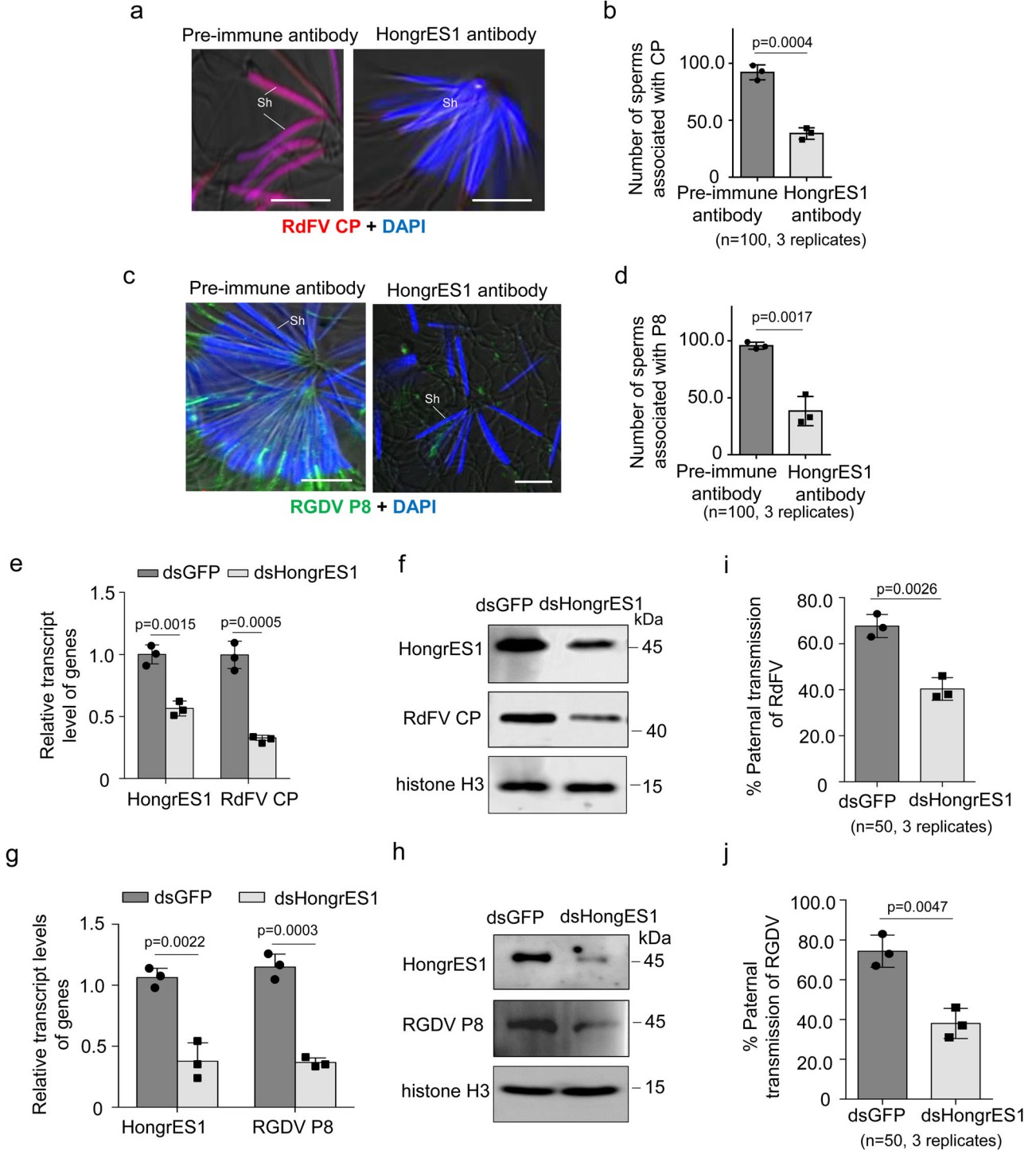

**Fig. 4 | HongrES1 mediates specific binding of RdFV or RGDV to sperms for paternal transmission by male _R. dorsalis_. a, b** Reduced CP-sperm association by pretreatment with HongrES1 antibody. Live sperms dissected from 30 virus-free testes were incubated with purified CP and pre-immune antibody or with purified CP and HongrES1 antibody, then stained with CP-rhodamine (red) and nuclear dye DAPI (blue) to indicate sperm head. Means (±SD) are shown from 100 sperms (3 replicates, two-tailed t-test). Sh, sperm head. Bars, 10 μm. **c, d** Reduced RGDV-sperm association by pretreatment with HongrES1 antibody. Live sperms dissected from virus-free testes were incubated with purified RGDV particles and pre-immune antibody, or with purified particles and HongrES1 antibody, then stained with P8-rhodamine (red) and DAPI (blue). Means (±SD) are shown from 100 sperms (3 replicates, two-tailed t-test). Sh, sperm head. Bars, 10 μm. **e–h** Effects of knockdown

of _HongrES1_ expression on RdFV or RGDV infection in reproductive organs in 30 dsHongrES1- or dsGFP-treated RdFV- (**e, h**) or RGDV- (**g, h**) positive males, as determined by RT-qPCR and western blot assays. Means (±SD) in **e** and **g** are shown from 30 male reproductive organs and represent three biological replicates (two-tailed t-test). The proteins were detected by using HongrES1, CP, P8 or histone H3 antibody in western blot assays. Data represent three biological replicates. **i, j** Paternal transmission rates of RdFV or RGDV, as determined by mating of dsHongrES1- or dsGFP-treated virus-positive males with virus-free females. Ten pairs of mating combination were performed for three biological replicates. Means (±SD) are shown from the 50 offspring of two mating combination, and represent three replicates (two-tailed _t_ test).

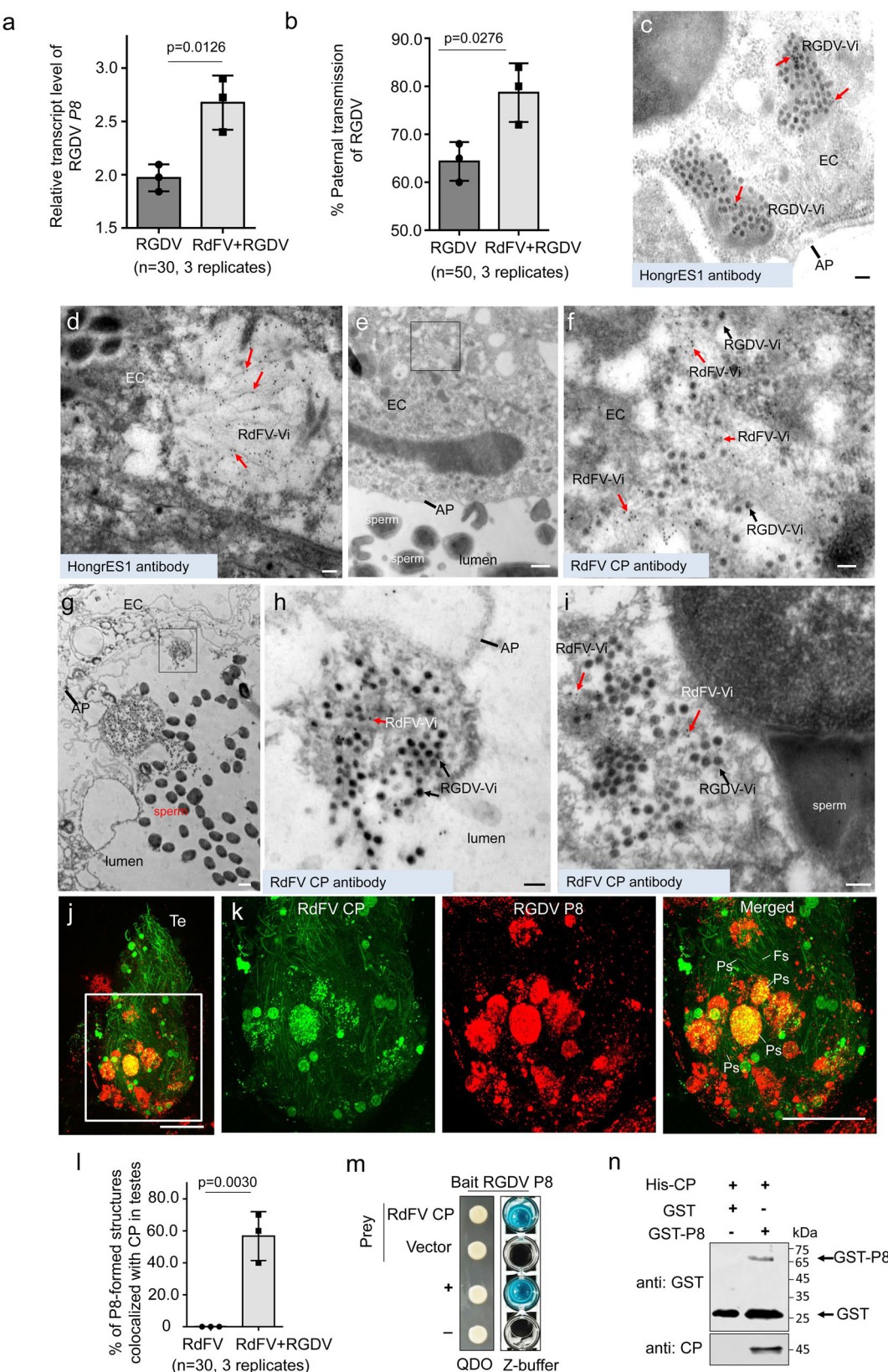

of *R. dorsalis*. RT-qPCR and western blot assays revealed that the conversion of PPO into PO was increased by RGDV infection in male reproductive system (Fig. 7b, c). However, the PO activity level in RGDV-positive male reproductive system was significantly lower than that in Gram-positive bacterium *Micrococcus luteus*-treated controls

(Fig. 7d). The knockdown of *PPO* expression by microinjection of synthesized dsRNAs targeting *PPO* (dsPPO) into RGDV-positive males decreased PO accumulation and activity, ultimately promoting viral infection in the male reproductive system (Fig. 7e, f). However, the knockdown of *HongrES1* expression by microinjection of dsHongrES1

**Fig. 5 | Direct interaction between RdFV CP and RGDV P8 mediates their simultaneous invasion of testes of male *R. dorsalis*. a** RT-qPCR assays showing the transcript levels of *P8* in 30 RGDV-positive or RdFV and RGDV co-positive male adults. Means (±SD) from three biological replicates (two-tailed t-test) are shown. **b** Paternal transmission rates of RGDV, as determined by mating of RGDV-positive or RdFV and RGDV co-positive males with virus-free females. Ten pairs of mating combination were performed for three biological replicates. Means (±SD) are shown from the 50 offspring of two mating combination, and represent three replicates (two-tailed t-test). **c, d** Immunoelectron microscopy showing the distribution of HongrES1 with RGDV and RdFV particles in testis epithelium. Immunoelectron microscopy showing the colocalization of RGDV and RdFV particles in testis epithelium (**e, f**) and sperm-accumulated testis lumen (**g–i**). Testes of leafhoppers were immunolabeled with HongrES1-specific IgG (**c, d**) or CP-specific IgG (**e–i**) as the primary antibody, followed by treatment with 15-nm gold particle- conjugated IgG as the secondary antibody. Red arrows indicate gold particles. **f** and **h** The enlarged images of the boxed areas in **e** and **g**, respectively. EC, epithelia cells. Vi, virions. AP, apical plasmalemma. Bars, 100 nm (**c, d, f, h, i**), and 500 nm (**e, g**). **j, k** Colocalization of CP and P8 in male testes, as determined by immunofluorescence microscopy. The dissected RdFV and RGDV co-positive testes were immunostained with CP-FITC (green) and P8-rhodamine (red). **k** The enlarged images of the boxed area in **j**. Te, testis. Ps, punctate structure. Fs, filamentous structure. Bars, 10 μm. **l** Mean percentage of P8-formed structures colocalized with CP in infected testes (*n* = 30). Means (±SD) represent three biological replicates (two-tailed t-test). **m** Y2H assay showing interaction of CP with P8. Transformants are labeled as follows: CP, pPR3-N-CP / pDHB1-P8; vector, pPR3-N/pDHB1-P8;+, positive control, i.e., pLargeT/p53; –, negative control, i.e., pDHB1/pRR3N. β-galactosidase assay was detected in Z buffer with X-Gal. QDO, SD/-Trp-Leu-His-Ade medium. **n** GST pull-down assay showing interaction of CP with P8.

into RGDV-positive males conversely increased PPO expression and PO activity, finally decreasing viral infection in the male reproductive system (Fig. 7g–i). Thus, RGDV infection activates a mild PO activity to induce antiviral melanization defense in the male reproductive system. Taken together, we demonstrate that RGDV infection triggers the increased expression of HongrES1 to suppress the conversion of PPO into active PO, which potentially modulates the balance between RGDV-induced melanization and effective RGDV infection in the male reproductive system.

## Discussion

A handful of studies describe the mechanistic basis of maternal arbovirus transmission in insect vectors and its role in viral persistence in nature[10,13,44]. On the other hand, the ancient obligate bacterial symbionts such as *Sulcia* and *Nasuia* in rice green leafhoppers can be vertically transmitted by female insects through eggs with 100% efficiency[16,20]. Arboviruses have apparently developed diverse strategies to enter oocytes for maternal transovarial transmission during long-term associations with insect vectors. For example, arboviruses can exploit the existing pathways to enter female insect oocytes, as used by vitellogenin, a major yolk protein precursor, and obligate bacterial symbionts such as *Sulcia* and *Nasuia*[16,19–21]. We have shown that RGDV in *R. dorsalis* vector and a symbiotic reovirus in psyllid host can exploit virus-induced tubules as the vehicles to enter the female oocytes[25]. Whether or how viruses develop other strategies to enter the oocyte remains elusive.

Currently, the mechanism of sperm-mediated paternal virus transmission by male insects in nature has been largely ignored. A key scientific question to be solved is whether insect sperm- and semen-specific components or symbiotic microorganisms could mediate paternal viral transmission. In mammals, HongrES1 is specifically expressed in the cauda epididymidis, secreted to the lumen for deposition on the sperm surface, and thus involved in sperm capacitation and maturation[29]. In this study, we identify HongrES1 homolog of *R. dorsalis* specifically present in the sperm surface and seminal fluid of males. We further reveal a new mode for RGDV and RdFV to coordinately activate and hijack HongrES1 for paternal transmission by males. Viral outer capsid proteins of RGDV and RdFV directly interact with HongrES1, and thus HongrES1 is able to gather at intact viral particles and the contact sites between sperm and virions. Pretreatment with HongrES1 antibody interferes with this interaction and strongly inhibits the binding of virions to the live sperms. The knockdown of HongrES1 expression effectively reduces paternal virus transmission. In this case, the HongeES1 on sperm surfaces can be regarded as the receptor for RGDV or RdFV. However, such virus-sperm binding does not lead to the invading of virions into the sperm cytoplasm. Our electron microscopy and paternal transmission experiments confirm that viral binding does not affect sperm morphological and functional characteristics. Finally, infected males are able to venereally transmit RGDV or RdFV to females during mating,

where the viruses are localized with the transferred sperms in the spermatheca. Such virus-decorated sperms in the female spermatheca finally move to the oviduct for fertilizing the eggs. Thus, HongrES1 serves as the receptors to mediate the direct binding of virions to the sperm surfaces and subsequent paternal virus transmission (Fig. 6g). Potentially, other arboviruses and symbiotic viruses might also have evolved to hitchhike with HongrES1 and its homologs on insect sperm surfaces for paternal transmission.

Phylogenetic analysis showed that the insect-specific RdFV, gramineous plants-infecting furoviruses, and insect-specific Hubei virga-like viruses belong to *Virgaviridae* family[38–40]. The close phylogenetic relationship between insect-infecting virgaviruses and their plant-infecting counterparts suggests that they might share the common virus origin(s)[39,45]. Plant virgaviruses are non-replicative in the insects[38,40,46,47], while insect-specific RdFV and Hubei virga-like virus are also non-replicative in the plants[39,45], suggesting the existences of host-specific barriers for virgavruses. In general, insect-specific symbiotic viruses such as *Nilaparvata lugens* reovirus and DcRV can be horizontally transmitted to their herbivorous insect hosts through plants[39,45,48,49]. Although vertical transmission rate of RdFV by *R. dorsalis* population is not 100%, rice plants might also serve as the passive vectors for the horizontal transmission of RdFV, finally ensuring the effective spread of RdFV population in nature. Recent metagenomic studies have found that insect-specific virgaviruses are widespread in insects[39], and may represent a major group of insect-specific paternally transmitted symbiotic viruses. More importantly, RdFV prolongs host adult longevity and promotes egg development, thereby benefiting the fitness of adults or their offspring. Thus, RdFV is beneficial to *R. dorsalis* population expansion, which explains the reason of mutualistic symbiosis between RdFV and *R. dorsalis* host.

Another important question to be solved is whether or how insect symbiotic microorganisms facilitate paternal arbovirus transmission. Some symbiotic viruses suppress the infection of arboviruses in insect vectors[1,3,4]. For example, cell-fusing agent virus negatively regulates the infection of both dengue and Zika viruses in mosquitoes[50], and palm creek virus suppresses the infection of West Nile virus in mosquitoes[51,52]. However, many symbiotic viruses are considered to benefit insect host fitness[51,52], and thus could affect the insect host' ability to acquire and transmit arboviruses. In this study, we show that RdFV is beneficial to insect host and offspring fitness, and thus RdFV existing in *R. dorsalis* population could facilitate vector competence for RGDV. We further reveal a direct interaction of RdFV and RGDV during their co-paternal transmission by *R. dorsalis*. The outer capsid proteins of these two viruses interact with each other, and RGDV spherical particles and RdFV filamentous particles are accumulated together in the testis epithelium, which are then together released into sperm-accumulated testis lumen via the apical plasmalemma. More importantly, HongES1 is associated with RGDV and RdFV particles in the testis epithelium and sperm-accumulated lumen, and gathers at the contact sites between sperm and virions. Thus, RdFV, RGDV and

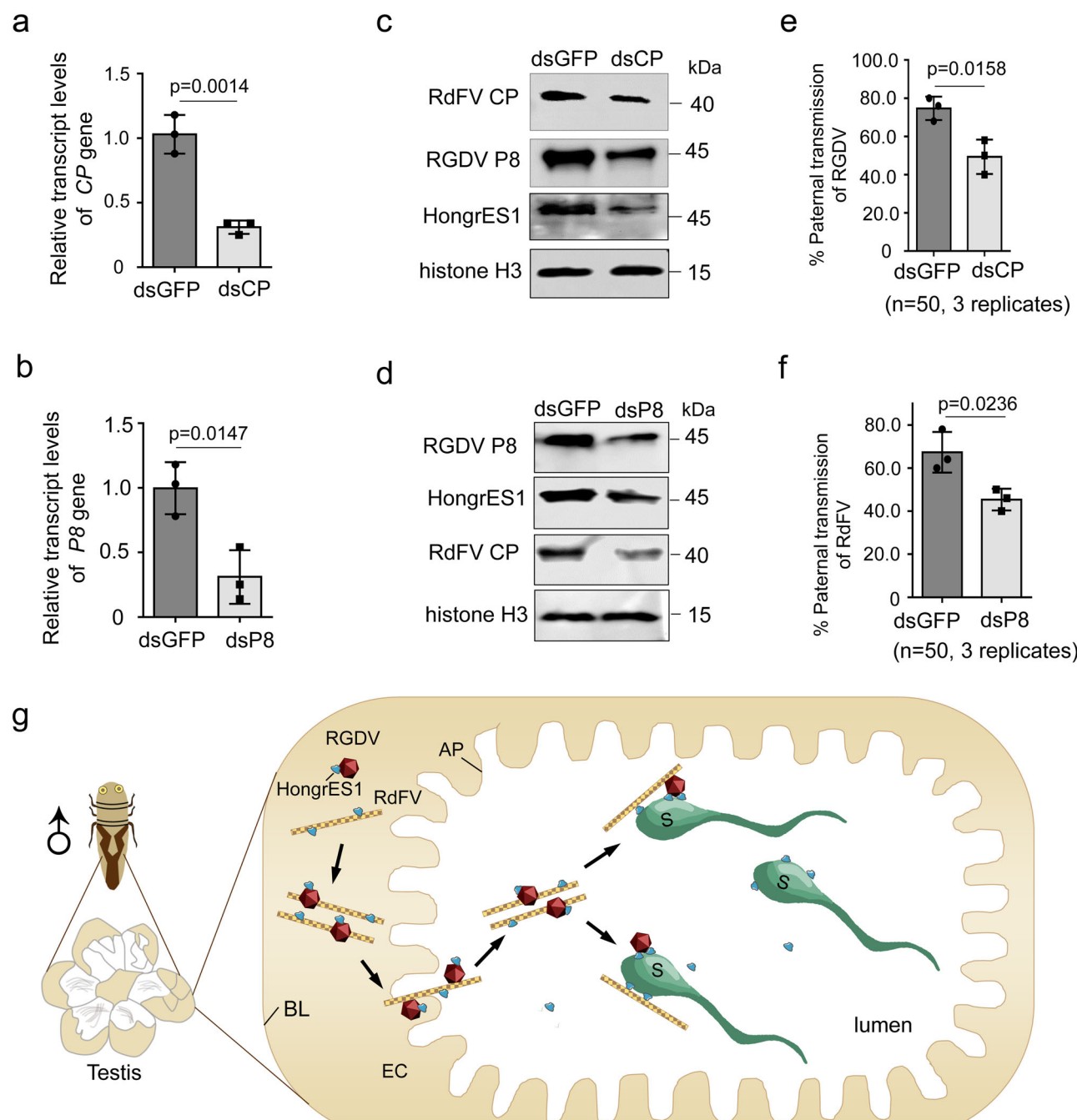

**Fig. 6 | Direct interaction between RdFV CP and RGDV P8 mediates their co-parental transmission by male *R. dorsalis*. a, b** RT-qPCR assays showing the transcript levels of RdFV *CP* (**a**) or RGDV *P8* (**b**) in 30 male reproductive organs of dsGFP-, dsCP- or dsP8-treated RdFV and RGDV co-positive male adults. Means (±SD) from three biological replicates (two-tailed t-test) are shown. **c, d** Western blot assays showing the protein accumulation levels of RdFV CP, RGDV P8 and HongrES1 in 30 male reproductive organs of dsGFP-, dsCP or dsP8-treated RdFV and RGDV co-positive male adults. The proteins were tested by using HongrES1, CP, P8 or histone H3 antibody. Bands for histone H3 demonstrate loading of protein. Data represent three biological replicates. **e, f** Paternal transmission rates of RGDV or RdFV, as determined by mating of dsCP-, dsP8- or dsGFP-treated RdFV and RGDV co-positive males with virus-free females. Ten pairs of mating combination were

performed for three biological replicates. Means (±SD) are shown from the 50 offspring of two mating combination, and represent three replicates (two-tailed *t* test). **g** Proposed model of a direct interaction of RdFV and RGDV during co-paternal transmission by *R. dorsalis*. RGDV and RdFV particles are accumulated together via P8-CP interaction to form the amorphous inclusion bodies in testis epithelium. Then the RGDV-RdFV complex release into sperm-accumulated testis lumen via passing through the apical plasmalemma. HongrES1 is recruited to RGDV or RdFV particles via interaction with P8 or CP. RGDV or RdFV particles are recruited on sperm via interactions among P8, CP and HongrES1, which facilitates RGDV and RdFV co-paternal transmission. AP, apical plasmalemma. BL, basal lamina. EC, epithelia cell. S, sperm.

HongES1 form a complex via the CP-HongrES1-P8 interaction. Taken together, RGDV and RdFV could activate and hijack HongrES1 for their simultaneous paternal transmission, potentially including the invasion of testis epithelium, release into sperm-accumulated lumen, and

attachment of sperm surfaces. However, whether RdFV virions could directly carry RGDV virions from testis epithelium into sperm-accumulated lumen remains elusive. In summary, our findings reveal a new mode for arboviruses and symbiotic viruses to cooperatively

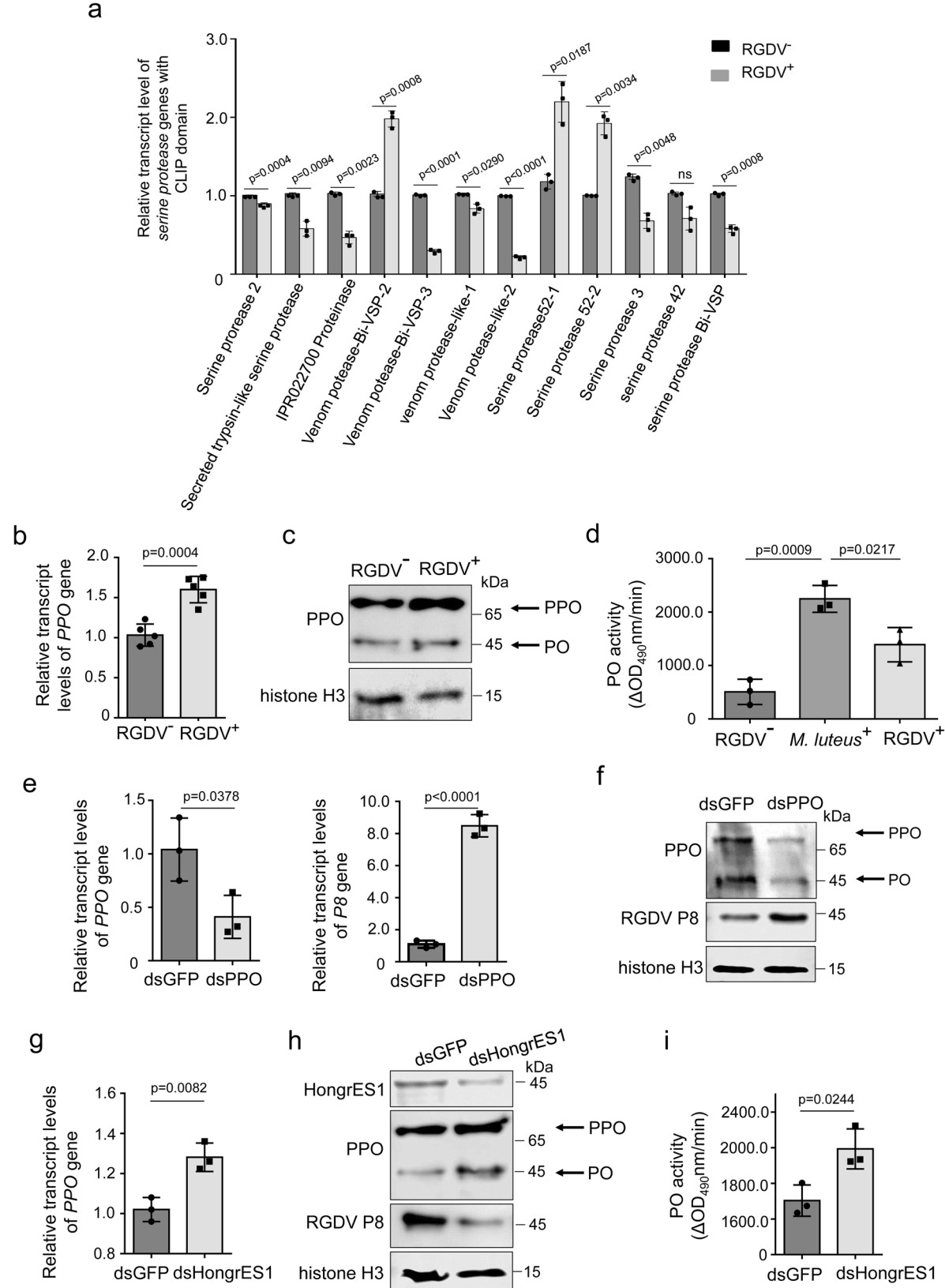

activate and hijack insect sperm-specific proteins for co-paternal transmission.

Over the past 40 years, viral disease caused by RGDV is always epidemic in the field in Southern China. During the winter months (November–March) in Guangdong, Southern China[53], RGDV-positive *R. dorsalis* population can keep up to two generations on the weeds such

as *Alopecurus aequalis*[24]. A male has a huge number of sperms and can mate repeatedly with females to enhance virus spread. Furthermore, RGDV does not affect the fitness of male-transmitted offspring[24,25,54], but causes significant fitness cost of female-transmitted offspring[24]. However, insect symbiotic RdFV is beneficial for adult host and offspring fitness, and thus facilitating vertical RGDV transmission. Thus,

**Fig. 7 | HongrES1 modulates PO activity to balance viral infection in male reproductive system of *R. dorsalis*. a** Effect of RGDV infection on the relative transcript level of *serine protease* genes containing clip-domain in male reproductive organs, as determined by RT-qPCR assay. **b, c** Effect of RGDV infection on the expression of PPO of male reproductive organs, as determined by RT-qPCR (**b**) and western blot assays (**c**). **d** PO activity in male reproductive organs of *M. luteus*-positive, and RGDV-positive or -free leafhoppers. **e, f** Effect of knockdown of *PPO* expression on RGDV infection in male reproductive organs, as determined by RT-qPCR (**e**) and western blot assays (**f**). **g, h** Effect of knockdown of *HongrES1* expression on PPO accumulation in male reproductive organs, as determined by RT-qPCR (**g**) and western blot assays (**h**). **i** Effect of knockdown of *HongrES1* expression on PO activity. Means (±SD) in **a, b, d, e, g** and **i** are shown from 30 male reproductive organs and represent at least three replicates. The protein accumulation levels of PPO, RGDV P8 or HongrES1 in **c, f** and **h** are shown from 30 male reproductive organs and represent three replicates. Bands for histone H3 demonstrate loading of equal amounts of protein.

the high efficiency for paternal RGDV transmission through males and the low efficiency for transovarial RGDV transmission via females simultaneously ensure effective viral transmission and insect fecundity. We deduce that sperm-mediated paternal RGDV transmission is a powerful type of vertical virus transmission by insect vectors, and plays a vital role in the efficient maintenance of RGDV during the cold seasons in the field. Many arboviruses such as La Crosse virus and Zika virus can also be paternally transmitted by male mosquitoes[12–15]. Potentially, insect sperm-specific proteins or symbiotic viruses might also facilitate paternal arboviruses transmission ensure viral long-term persistence in nature.

Finally, we reveal a new function of HongrES1, like other members of the serpin family, in mediating insect antiviral melanization defense. HongrES1 has the serine proteinase inhibitor activity to modulate PO activity to benefit viral infection in insect male reproductive system. Once pathogen-associated molecular patterns are recognized, the PPO is converted to active PO via serpin-regulated serine proteases, while PO catalyzes the formation of melanin to immobilize microbes[55]. We found that the increased HongrES1 expression during RGDV infection effectively suppresses the expression of clip-domain serine proteases and the subsequent conversion of PPO to the active PO, ultimately producing a mild PO activity in the male reproductive system. Thus, viral infection potentially activates HongrES1 expression to modulate the balance of virus-induced melanization and effective viral infection, guaranteeing the normal role of HongrES1 in sperm maturation and paternal viral transmission.

## Methods

### Identification, validation, and analyses of RdFV genome sequence

*R. dorsalis* adults were collected from rice fields in Luoding city, Guangdong Province, China from 2018, and propagated on TN-1 rice seedlings in cages at $25 \pm 1\,°C$ with $75 \pm 5\%$ relative humidity and 16-h light/8-h dark. High-throughput sequencing was used for identification of insect-specific viruses from this *R. dorsalis* population. Total RNA was extracted from *R. dorsalis* adults in TRIzol reagent according to the manufacturer's instructions. High quality RNAs were selected for construction of small RNAs and transcriptomic libraries, which were sequenced using an Illumina NovaSeq 6000 platform in Novogene Co., Ltd, China. Low-quality reads and adapter sequences were removed from the raw reads to obtain clean reads. The clean reads were assembled, and the assembled contigs were analyzed using BLASTx searches in the nonredundant protein database available in NCBI. BLAST results were then checked carefully to screen potential viral sequences.

A new positive-sense single-stranded RNA virus was screened and identified using RT-PCR assays, and then named as RdFV. The primers used in RT-PCR assays were shown in Supplementary Table 1. Full length of RdFV genome sequences were analyzed by NCBI ORF finder online. Phylogeny was analyzed based on comparison of RdRp amino acid sequences with counterparts in insect or plant virgavirus lineages using Bayesian inference in MrBayes 3.2.6 under the rtREV+F + G4+I model[56]. This model was determined using the Bayesian information criterion by ModelFinder[57]. Markov chains were run for 2,000,000 generations, sampling every 100 generations. The sufficient sampling and parameter convergence were checked using Tracer 1.71, after

discarded the first 25% samples as burn-in. The Bayesian 50% majority rule consensus tree was visualized in FigTree 1.4.4.

### Insects, viruses, and antibodies

Our preliminary experiments using RT-PCR assay showed that about 80% of male (♂) or female (♀) *R. dorsalis* population ($n = 100$, 3 replicates) reared under controlled greenhouse conditions had the transcript of RdFV *CP* (Fig. 1B). To establish the RdFV-positive or free leafhopper colony, pairs of one female and one male were individually kept in glass tubes containing one rice seedling to lay eggs. The parents were tested for RdFV using RT-PCR assays, and the offspring produced by RdFV-positive or free parents were reared to establish RdFV-positive or free population. The primers used in RT-PCR assays were shown in Supplementary Table 1.

Rice plants infected with RGDV isolates were also originally collected from Luoding city, Guangdong Province, China and maintained on rice plants via transmission by *R. dorsalis*. To obtain RGDV-positive or RdFV and RGDV co-positive *R. dorsalis* population, the 2th instar nymph of RdFV-free or positive leafhoppers were fed on RGDV-infected rice plants for 2 day and then transferred to healthy rice seedlings. At 14-day post-first access to diseased plants (padp), the presence of RdFV or RGDV was identified using RT-PCR assays. The offspring produced by RGDV-positive or RdFV and RGDV co-positive parents were reared to establish RGDV-positive or RdFV and RGDV co-positive population. The primers used in RT-PCR assays were shown in Supplementary Table 1.

Rabbit polyclonal antibodies against RdFV CP, HongrES1, RGDV P8 and PPO were prepared by Genscript Biotech Corporation, Nanjing, China. The process was approved by the Science Technology Department of Jiangsu Province of China. Specific IgG against RdFV CP or RGDV P8 was conjugated to rhodamine to generate CP-rhodamine or P8-rhodamine. Specific IgG against HongrES1, RGDV P8 or RdFV CP was conjugated to fluorescein isothiocyanate (FITC) to generate HongrES1-FITC, P8-FITC, or CP-FITC. Mouse monoclonal antibody against GST was purchased from Transgene Biotech (HT601). The actin dyes Alexa Fluor 647 Phalloidin, and the nuclear dye 4',6-diamidino-2-phenylindole (DAPI) were purchased from Thermo Fisher Scientific (A22287, 62248). Rabbit polyclonal antibody against histone H3 was purchased from Abcam (ab1791).

### RT-qPCR assays

To analyze the expression levels of genes of viruses or *R. dorsalis*, the total RNAs were extracted from tissues or whole bodies using TRIzol reagent (Thermo Fisher Scientific, 15596026). The relative quantification of genes was tested by RT-qPCR assay using the 2 × RealStar Fast SYBR qPCR Mix (High ROX; Genstar, A303) in the QuantStudio 5 Real-Time PCR System (Thermo Fisher Scientific). The primers used in RT-qPCR assays were shown in Supplementary Table 2. The transcript level of the housekeeping gene *elongation factor 1 alpha* (*EF1α*) of *R. dorsalis* (Genbank accession number: AB836665) served as the internal reference for the normalization of gene expression levels. Relative gene expression levels were estimated by the $^{-\Delta\Delta}$CT (cycle threshold) method[24]. The CT value of 35 was served as a cutoff value, and the CT values lower than 35 were considered as effective data. The experiments were replicated at least three times, and a pool of 30 leafhoppers was used for each replicate.

## Immunofluorescence microscopy

For visualizing RdFV infection to different tissues of *R. dorsalis*, the alimentary canal, salivary gland, and male or female reproductive organs were dissected from 30 RdFV-free or -positive *R. dorsalis* leafhoppers. The samples were fixed in 4% (v/v) paraformaldehyde in PBS for 2 h, and then permeabilized in 0.2% (v/v) Triton-X for 1 h. The samples were then immunolabeled with RdFV CP-rhodamine (0.5 μg/μl) and the actin dye Alexa Fluor 647 Phalloidin (0.1 μg/μl). Immunostained tissues were visualized using a Leica TCS SPE inverted confocal microscope.

For visualizing the association of HongrES1 with RdFV or RGDV in the male reproductive system, the reproductive organs were dissected from 30 RdFV-free, RdFV-positive, or RdFV and RGDV co-positive males. The samples were fixed, permeabilized, immunolabeled with HongrES1-FITC, CP-FITC, CP-rhodamine, or P8-rhodamine (0.5 μg/μl), and then processed for immunofluorescence microscopy.

For visualizing virus or HongrES1 association with sperms, mature sperms were excised from the testes of 30 RdFV-free, RdFV-positive, or RdFV and RGDV co-positive males, and then smeared on poly-lysine-treated glass slides. The sperms were successively fixed, permeabilized, immunolabeled with CP-rhodamine, P8-rhodamine, P8-FITC, HongrES1-FITC (0.5 μg/μl), stained with DAPI (2.0 μg/ml), and then processed for immunofluorescence microscopy.

## Transmission electron microscopy

For electron microscopy, the midgut or reproductive organs of RdFV-free, RdFV-positive or RdFV and RGDV co-positive male or female adults, or the spermatheca of virus-free females at different days after mating with RdFV-positive males were excised, fixed with 2% (v/v) glutaradehyde and 2% (v/v) paraformaldehyde in PBS for 2 h at room temperature, and then postfixed with 1% (w/v) osmium tetroxide in PBS for 1 h at room temperature. The fixed samples were dehydrated in grade series of ethanol up to 100% and embedded in Spurr's resin (SPI Ltd). Samples were sectioned on an ultramicrotome (Leica) with a diamond knife. The ultrathin sections were observed with a transmission electron microscope (H-7650; Hitachi).

For immunoelectron microscopy, the samples were fixed with 2% (v/v) glutaradehyde and 2% (v/v) paraformaldehyde in PBS for 2 h at room temperature. Fixed samples were dehydrated through a graded ethanol series at −20 °C and embedded in LR gold resin (Bioscience). Polymerization was allowed to proceed for 72 h at −20 °C. Samples were sectioned on an ultramicrotome (LKB Nova) with a diamond knife and the ultrathin sections were then immunolabeled with HongrES1-, CP- or P8-specific IgG (0.5 μg/μl) as the primary antibody, and then treated with goat anti-rabbit IgG conjugated with 15-nm diameter gold particles (0.5 μg/μl; Abcam) as the secondary antibody. Then the samples were examined under the transmission electron microscope.

## Vertical transmission of RdFV or RGDV

Four mating combinations from the lab-reared *R. dorsalis* colony were conducted as follows: (i) infected virgin female × infected male; (ii) infected virgin female × uninfected male; (iii) uninfected virgin female × infected male; and (iv) uninfected virgin female × uninfected male (Supplementary Table 3). In each combination, 50 newly emerged females and 50 newly emerged male adults mated one to one in glass tubes containing one rice seedling for 3 days. The rice seedlings in glass tubes were replaced into new ones each day to avoid viral acquisition from rice plant. The females and males were then tested by RT-PCR assay to further confirm uninfected or infected parents. The eggs laid by these four combinations were harvested at 7-day post oviposition by dissecting rice seedlings, and were individually placed on a piece of water-soaked filter paper in petri dishes at 25 ± 3 °C[54]. After eggs eclosion, offspring were individually fed on new rice seedlings, and the presence of RdFV or RGDV were also tested by RT-PCR assays. The primers used in RT-PCR assays were shown in

Supplementary Table 1. Three independent biological replicates of each mating combination were conducted and analyzed.

## Effect of RdFV on *R. dorsalis* fitness

Fifth-instar nymphs of *R. dorsalis* from RdFV-free or -positive *R. dorsalis* colony were individually reared with one healthy rice seedling in a glass tube. The growth of each leafhopper was monitored at 12-h interval until the end of adult life to measure the adult longevity. Longevity of 50 RdFV-free or positive female or male adults was analyzed. The entire experiment was replicated three times.

To examine the effect of RdFV on *R. dorsalis* offspring, two mating combinations from the lab-reared *R. dorsalis* colony were conducted as follows: (i) infected virgin female × infected male; and (ii) uninfected virgin female × uninfected male (Supplementary Table 3). In each combination, 10 newly emerged females and 10 newly emerged male adults mated one to one in glass tubes containing one rice seedling for 3 days. At 7-day post oviposition, the eggs were collected, and the number of eggs was recorded. Red eyespots on eggs served as an indicator of embryonic development. The egg hatching rate of each combination at 9-, 11-, 13- and 15-day post oviposition was calculated as number of eggs with eyespots/total 50 eggs. Fifty eggs with red eyespots were randomly collected from each combination was measured for the length of each egg using an anatomical lens and imaging equipment (Nikon SMZ18). Fifty eggs were randomly collected from each combination was monitored at 12-h interval until nymph emergence to determine the duration of egg development. Three independent biological replicates of each mating combination were conducted and analyzed.

## Y2H assay

To examine the interaction among RdFV CP, HongrES1, and RGDV P8, the DUALmembrane starter kit (Dualsystems Biotech, P01201-P01229) was used according to the manufacturer's instructions. The full-length ORFs of RdFV CP and RGDV P8 were separately inserted into bait vector pDHB1 to generate pDHB1-CP and pDHB1-P8, and the full-length ORFs of HongrES1, RdFV CP, RSMV G, and RDV P8 were separately inserted into prey vector pPR3-N to generate pPR3-N-HongrES1, pPR3-N-CP, pPR3-N-G and pPR3-N-P8 (RDV). The bait and prey were co-transformed the yeast strain NMY51. The pLargeT/p53 interaction served as a positive control, and the pDHB1/pRR3N served as a negative control. Transformants were subsequently screened on the QDO (SD/-Trp-Leu-His-Ade/X-Gal) culture medium, and β-galactosidase activity was detected in Z buffer with X-Gal. The primers used in Y2H were shown in Supplementary Table 1.

## GST pull-down assay

The ORFs of RdFV *CP*, *HongrES1* and RGDV *P8* were separately cloned into the pGEX-4T-3 vector to construct plasmids expressing the GST fusion protein as baits (GST-CP, GST-HongrES1 and GST-P8, respectively). The full-length ORFs of *HongrES1*, RGDV *P8* and RdFV *CP* also were separately cloned into the pEASY-Blunt E1 Expression Vector (Transgen Biotech, CE111−01) to construct plasmids expressing the His fusion protein as preys (His-HongrES1, His-P8 and His-CP, respectively). The recombinant proteins fused with GST tag and GST protein were respectively expressed in *E. coli* strain BL21. Lysates were then subsequently incubated with glutathione-Sepharose beads (GE Healthcare, 17-0756-01) and expressed RdFV CP, HongrES1 or RGDV P8, respectively. The eluates were analyzed by western blot assays using antibodies against GST-tag, HongrES1, RGDV P8 and RdFV CP (0.5 μg/μl). The primers used in GST pull-down assays were shown in Supplementary Table 2.

## Expression analysis of HongrES1 in male reproductive system of *R. dorsalis* during RdFV or RGDV infection

To analyze the expression levels of HongrES1 in different tissues of *R. dorsalis*, the whole body, alimentary canal, reproductive organs and

salivary gland were dissected from 30 RdFV-free males or virgin females at 5-days post eclosion. The relative expression of *HongrES1* in different tissues was detected by RT-qPCR assays. To verify the expression patterns of HongrES1, the total proteins were extracted from various tissues of 30 RdFV-free males or females, and then analyzed by western blot assays. Antibodies against HongrES1 and histone H3 (0.5 µg/µl) served as the primary antibodies, and goat anti-rabbit IgG-peroxidase (0.5 µg/µl) served as the secondary antibody.

We also detected the effects of RdFV or RGDV infection on the expression levels of HongrES1 in the male reproductive system. The reproductive organs were dissected from 30 RdFV-free, RdFV-positive, or RGDV and RdFV co-positive males. The relative expression of *HongrES1* was detected by RT-qPCR assays. In the corresponding western blot assay, antibodies against HongrES1, RdFV CP, RGDV P8, and histone H3 (0.5 µg/µl) served as the primary antibodies, and goat anti-rabbit IgG-peroxidase (0.5 µg/µl) served as the secondary antibody. At least three biological replicates were performed.

### Neutralizing virus-sperm binding
To detect the binding of RdFV CP with sperms in vitro, His-tag-fused CP was expressed in *Escherichia coli* strain Rosetta, and the proteins were purified using nickel-nitrilotriacetic acid resin (Qiagen). Sperm smears collected from testes of RdFV-free *R. dorsalis* were successively incubated with the purified CP (0.5 µg/µl) for 1 h, smeared on poly-lysine-treated glass slides, immunolabeled with CP-rhodamine (0.5 µg/µl), stained with DAPI (2.0 µg/ml), and then processed for immunofluorescence microscopy.

In neutralization experiments to test the direct interaction between RdFV CP and HongrES1, mature sperms excised from the testes of RdFV-free *R. dorsalis* were pre-incubated for 30 min with preimmune antibody (0.5 µg/µl) or HongrES1 antibody (0.5 µg/µl), and then the in vitro CP-sperm binding experiment was performed as described above.

In neutralization experiments to test the direct interaction between RGDV particles and HongrES1, mature sperms excised from RGDV-free leafhoppers were pre-incubated for 30 min with pre-immune antibody (0.5 µg/µl) or HongrES1 antibody (0.5 µg/µl), incubated with the purified RGDV particles (1.0 µg/µl) for 1 h, smeared on poly-lysine-treated glass slides, immunolabeled with P8-FITC (0.5 µg/µl), stained with DAPI (2.0 µg/ml), and then processed for immunofluorescence microscopy.

### Effect of synthesized dsRNAs on viral infection in male reproductive system and paternal transmission
RNA inference was performed to knock down the expression of related genes. The T7 promoter with the sequence 5′-ATTCTCTA-GAAGCTTAATACGACTCACTATAGGG-3′ was added to the forward and reverse primers at the 5′ terminal to amplify a region of ~500–800 bp of *HongrES1*, RdFV *CP*, RGDV *P8*, or *GFP* gene (Supplementary Table 1). The PCR products were used for the synthesis of dsRNAs targeting *HongrES1* (dsHongrES1), RdFV *CP* (dsCP), RGDV *P8* (dsP8), or *GFP* (dsGFP) according to the protocol for the T7 RiboMAX Express RNAi System kit (Promega, P1700).

To test the knockdown of *HongrES1* expression on RdFV or RGDV infection in male reproductive systems, newly emerged male adults of RdFV-positive or RGDV-positive *R. dorsalis* population were microinjected with dsHongrES1 or dsGFP (approximately 200 ng/leafhopper) using a Nanoject II Auto-Nanoliter Injector (Spring), and then transferred to healthy rice seedlings. To test the knockdown of RdFV *CP* or RGDV *P8* expression on viral infection and HongrES1 accumulation in male reproductive system, newly emerged male adults of RdFV and RGDV co-positive *R. dorsalis* population were microinjected with dsCP, dsP8 or dsGFP (~200 ng/leafhopper), and then transferred to rice seedlings. For each treatment, approximate 100 insects were microinjected, and three replicates were performed.

The male reproductive organs were dissected to test the expression levels of RdFV CP, RGDV P8, or HongrES1 using RT-qPCR and western blot assays. A pool of 30 dsRNA-treated males was used for each replicate, and the experiment was conducted in three replicates for RT-qPCR assays. The total proteins from reproductive organs of 30 dsRNA-treated males were analyzed for the protein levels in western blot assays by using HongrES1-, CP- or P8-specific IgG (0.5 µg/µl). Experiment was conducted in three replicates in western blot assays. To determine the effect of dsHongrES1, dsCP or dsP8 treatment on paternal transmission of RdFV or RGDV, one dsHongrES1-, dsCP-, dsP8- or dsGFP-treated RdFV- or RGDV-positive male mated with one virus-free virgin female in a glass tube containing a rice seedling for 3 days (Supplementary Table 3). Ten pairs were performed for each treatment. The males were then tested for presence of RdFV or RGDV, and females were left in the tubes for oviposition. The offspring of each mating combination were tested for presence of RdFV or RGDV by RT-PCR assays. The primers used in RT-PCR assays were shown in Supplementary Table 1. The experiment was conducted in three replicates.

### Effect of RdFV on RGDV infection in male reproductive system of *R. dorsalis* and subsequent paternal transmission
To test the effect of RdFV on RGDV infection, the whole bodies dissected from 30 RGDV-positive or RGDV and RdFV co-positive *R. dorsalis* adults were individually analyzed by RT-qPCR assays to determine the relative transcript levels of RGDV *P8*. To detect the effect of RdFV on paternal RGDV transmission, one virgin male co-infected by RGDV and RdFV or singly infected by RGDV mated with one virus-free virgin female for 3 days (Supplementary Table 3). Fifty pairs were performed for each mating combination. After mating, males were tested in RT-PCR assays to determine the presence of RGDV or RdFV. The offspring of each mating combination were tested for RGDV by RT-PCR assays. The primers used in RT-PCR assays were shown in Supplementary Table 1.

### Effect of RGDV infection on the conversion of PPO to active PO in *R. dorsalis*
The reproductive organs were individually dissected from the newly emerged male adults of RdFV and RGDV co-positive *R. dorsalis* population, and the relative transcript levels of *clip-domain serine protease* genes and *PPO* were examined by RT-qPCR assays. The male reproductive organs were also examined to determine the conversion of PPO to PO in western blot assays using PPO and histone H3 antibodies (0.5 µg/µl). A pool of 30 RGDV-positive males was used for each replicate in RT-qPCR and western blot assays, respectively. The experiment was conducted in at least three replicates for RT-qPCR and western blot assays. To analyze effect of RGDV infection on PO activity, the reproductive organs dissected from approximate 100 newly emerged males were homogenized with the His-Mg buffer (0.1 M histidine, 0.01 M MgCl$_2$, pH 6.2) buffer in liquid nitrogen. The supernatant was gently mixed with 1 mM dopamine in 10 mM Tris-HCl buffer (pH 8.0) in a 96-well plate at room temperature for 5 min. Enzyme activity was measured using the phenoloxidase kit (Geruisi, G0146W) according to the manufacturer's protocol. To analyze the effect of *M. luteus* infection on PO activity, freeze-dried *M. luteus* was dissolved in water, and then microinjected in dose of ~23 ng/leafhopper into newly emerged males. At 24-h post microinjection, the reproductive organs of approximate 100 RGDV-infected or *M. luteus*-treated males were dissected and tested for PO activity.

We then tested the effect of knockdown of *PPO* or *HongrES1* expression on PO activity and RGDV infection. The newly emerged male adults of RdFV and RGDV co-positive *R. dorsalis* population were microinjected with dsGFP, dsPPO or dsHongrES1 (~200 ng/leafhopper). The male reproductive organs of these tested leafhoppers were individually collected and dissected for RT-qPCR and western blot assays to determine the effect of dsRNAs on the expression levels of

HongrES1, PPO, or RGDV P8, and the conversion of PPO to active PO, as well as PO activity. A pool of 30 males was used for each replicate in RT-qPCR and western blot assays, respectively. A pool of 100 males was tested for each replicate in PO activity. The experiment was conducted in three replicates for RT-qPCR and western blot assays, as well as PO activity tests.

### Statistical analyses and reproducibility
To analyze the probability of discrepancy occurrence in a specific range, all quantitative data presented in the text and figures were analyzed with unpaired two-tailed t-test in GraphPad Prism 7 software (GraphPad Software, San Diego, CA, USA).

All micrographs of electron, immunoelectron, immuno-fluorescence and transmitted light were representative of the results of at least 3 biological experiments. Data showed in GST pull-down assays were representative of 3 biological experiments.

### Reporting summary
Further information on research design is available in the Nature Portfolio Reporting Summary linked to this article.

## Data availability
The authors declare that all data supporting the findings of this study are available in the manuscript and its Supplementary Information files. Source data are provided with this paper. The nucleotide sequence of RdFV generated in this study were deposited in the Genbank under accession code OP326514. Full uncropped scans of presented blots are provided in the Supplementary Information File. Source data are provided with this paper.

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

## Acknowledgements

We thank Professor Fangluan Gao of Fujian Agriculture and Forestry University for assistance with phylogeny analyses of RdFV RdRp and construction of Bayesian phylogenetic tree. This work was supported by the National Natural Science Foundation of China (Nos. 31730071, 31920103014 and U21A20221).

## Author contributions

T.W. and Q.C. designed the research; J.W., Q.L., R.Z., Y.C., X.W., H.W., J.Z., Y.D., and D.J. performed the research. J.W., Q.L., and R.Z. contributed equally to this work. H.W., D.J., W.Z., D.T, T.W., and Q.C. analyzed the data; T.W. and Q.C. wrote the paper. All authors read and approved the manuscript.

## Competing interests

The authors declare no competing interests.
