## [Peer Review File · Nature Communications]

REVIEWER COMMENTS

Reviewer #2 (Remarks to the Author):

In this study, the authors identify a new sperm-specific serpin protein HongrES1 of leafhopper *Recilia dorsalis* for mediating paternal transmission of a rice reovirus and a novel symbiotic virus of the Kitaviridae family. This work is a continuation of the previous work by the same group where they that major outer capsid protein P8 of RGDV interacts with heparan sulfate proteoglycan (HSPG) to facilitate sperm-mediated paternal virus transmission. HongrES1 serves as the receptors to mediate the direct binding of virions to leafhopper sperm surfaces and subsequent paternal transmission via interaction with both viral capsid proteins. The study is well-conducted and the findings reveal a new mode for symbiotic viruses to synergistically hijack insect sperm-specific proteins for paternal transmission without disturbing sperm functions.

Overall, the manuscript is well-written and the figures are well-explained. Few comments below

Figure 1: It will be helpful to show the immunoelectron microscopy and Western blot analysis of the female RO as well for comparison

Figure 2 and 3 : Include quantitative data (Image J) from 3 or more independent expts for IFA images.

Figure 4: Including rescue experiments to support this data will strengthen the claim

Figure 5K: does dsCP affect the expression of HongrES1?

IFA Co-localization indicates that majority of RGDV staining (red) doesn't co-localize with RdFV staining (green). To resolve this concern, a co-localization quantitative examination will be helpful.

Discussion: Maternal transmission may still be possible using another mechanism and hasn't been fully examined. This should be clarified.

Authors explain that sperm function isn't affected by the two viruses. Has there been any morphological and functional analysis of the sperm?

Reviewer #3 (Remarks to the Author):

This paper presents investigations of a largely unexplored mechanism for vertical arbovirus transmission, using viral binding to insect sperm receptors. The topic is very interesting and of significant importance as there is very limited knowledge on virus long term viral persistence, independent on availability of other viral hosts. I appreciate the investigations and comprehensive explanations, but the text lacks some important details of the work.

Introduction

The statement “scarcely affects insect fitness” is repeated twice (L41 and L47) but is only based on one reference, which is self-citation. The statement needs more support or less boldly expressed, and not repeated twice.

L49: Change to: “...arbovirus transmission may have evolved as a preferred mode of vertical transmission during the long-term virus-vector interaction.”

L51: The sentence starts with “..insect symbiotic viruses” but in the following sentence you give ticks as an example. Use arthropod symbiotic viruses or remove ticks as an example as they are not insects.

L63: Do you mean “relatively complicated” (missed a d)? Specify, what do you mean with complicated components.

Results

You have a lot of information in the figures that is lacking from the main text. Please add sample sizes, number of replicates and means, SDs, significance details etc to the main text (results/methods).

L91: How did you identify this virus and from where? I cannot find this as a reference, nor in any details in the methods.

L96: How many were tested?

L101: by how many days?

L135: plays

L139: To use the word titers is incorrect as you haven't titrated the virus and measured functional virions, you have only measured viral RNA with RT-PCR. It is an important distinction.

Discussion

You have shown that the virus particles associate with the receptor but you don't discuss potential infection and replication. Could you elaborate on this discrepancy in the discussion.

L226-227 repeats L221-222 and needs editing.

L242-244 "We anticipate that other....". Why do you anticipate this? Any references pointing to the same conclusion? Otherwise it is better to write "Potentially, other arboviruses..."

Methods

Please provide more details on insect collection. When were insects and plants collected from the field? How was infection confirmed? Which greenhouse conditions?

You use a lot of RT-PCR but there are no details of this PCRs. Please provide references and/or primers/cutoff values used.

L310-311: I don't understand this statement. Do you mean they were tested for the presence of RdFV to confirm infection? Or did you not know if they were infected when you set up the mating pairs?

L312-313: Here you write that offspring of crossing-pairs were raised to adults and then tested for RdFV. In the figure legends (L605-606, L667, L705) you write that eggs were collected. Which is correct?

L326: give reference to method.

L355: Replace detected with tested: "the bodies were tested for"

L361; Replace detected with tested: "males were tested for"

Figures:

In general, the figures are too small, and with a too long legend text, making them unnecessarily difficult to follow. I recommend to divide them into several separate figures and condense the legend texts.

L679, L699: "Approximate 30...". How can numbers of tested in RT-qPCR be approximate?

S1: Why do you use neighbor-joining and not maximum likelihood or Bayesian methods? Support numbers not clearly visible and are also very low.

S2: What does the figure of the morphology of eggs say? To me they look the same.

Legend of blue and red markings is incomplete.

Reviewer #4 (Remarks to the Author):

The study system presented by Wan et al. provides new pieces of molecular (functional proteomics, gene silencing) and ultrastructural cellular evidence to support one mechanism underlying the previously reported phenomenon of paternal transmission of a leafhopper-transmitted plant reovirus (rice gall dwarf virus, RGDV) via sperm to ova of female leafhoppers (rice green leafhopper, *Recilia dorsalis*). In addition, the authors discovered a new filamentous virus (RdFV) associated with the leafhopper in their lab colony (80% incidence). The authors conducted various mating experiments to test hypotheses to address the capability of this viral associate to be sexually-transmitted, if this viral associate affects transmission efficiency of the plant reovirus to leafhopper progeny and plants, and in parallel, performed several molecular and cellular analyses to determine if this new vector-associated virus (referred to 'symbiont' and 'endosymbiont' in the manuscript) utilizes the same mechanism of sperm attachment and transport to ova as the plant reovirus. The foundation for the study was well established previously by the authors and by others examining paternal transmission by vector transmitted vertebrate viruses (mosquito) and other virologists studying other insect reoviruses and their nonstructural protein associated with sexual transmission (cited in references: Chen et al., 2019). Analogies with mammal sperm provided the focus on an insect ortholog of a serine protease inhibitor (serpin: HONGrES1) associated with sperm surface structure and function, which also formed the basis that this serpin may modulate (dampen) insect innate immunity: reduced phenol oxidase production and thus reduced melanization.

The methodologies used to generate evidence in support of the sperm-mediated mechanism (protein-protein interactions, transmission microscopy and immunolocalization of target proteins) of male-transmission are gold-standard methods in virology and microbe-host interactions, and the experimental design includes independent, replications of experiments and sufficient numbers of experimental units (insects) to provide robust analyses. While the findings regarding the shared mechanism of paternal transmission of the reovirus (RGDV) and symbiont (RdFV) are solid, there are some major concerns that need to be addressed explicitly to improve the quality of the study.

1. The authors conclude that the two viruses interact synergistically to 'hijack' sperm proteins for transmission. The use of the word 'synergy' and its derivatives in the title and results/discussion/figure legend text are not supported by the design of the experiment nor the analysis of the data. Several readers, including this reviewer, define synergy as: the combined effect of two entities (RGDV and RdFV) is significantly greater than the sum of their individual effects on a third entity (response of third entity). Since the authors are interested in how one virus (RdFV) affects the transmission of the other virus (RGDV), it is impossible to demonstrate synergy since one cannot show the single effect of RdFV (in absence of RGDV) on transmission of RGDV; RdFV must always be with RGDV to have an effect on RGDV transmission. Just for illustrative purposes, suppose sperm fitness was the response variable (3rd entity = sperm), then a synergistic or additive or antagonistic effect could be demonstrated by: the combined effect of [RGDV infection (entity A) and RdFV infection (entity B)] on sperm fitness is greater/equal/less than sum of RGDV effect alone on fitness and RdFV effect alone on fitness, and there are statistical methods to distinguish additive from synergistic effects. In addition, the authors' interpretation of 'synergy' was based on results displayed in Figure 5 and perhaps Fig 3b(?): this reviewer does not interpret synergy based on the data in this figure. In figure 3b, one would reasonably interpret these data as RdFV enhancing transmission of RGDV, but one cannot conclude synergy between the viruses - statistical support for synergy is not possible without measuring a response of single entities, and in fact, it appears that RdFV increases rate of RGDV transmission from ~62% to ~79%. How can one say this is synergistic, additive or marginal? Please address this concern explicitly in the text - this reviewer does not define synergy between 2 entities as the effect of one entity on the other entity.

2. The authors presumed that RdFV (symbiont) facilitates or enhances sperm infection and paternal transmission by the plant reovirus (RGDV). Is the alternative possible: that RGDV facilitates, enhances sperm entry, and paternal transmission by the symbiont RdFV? The authors made a statement in the discussion (line 260) that 'paternal transmission route that is established for the more ancient RdFV..'. The authors made no effort to determine the evolutionary history and age (molecular clock analysis) of RdFV vs RGDV. There is in fact, no discussion about this new virus and readers would be interested in knowing more about the membership (taxon) of this virus. The authors must provide evidence (molecular clock analysis or other virus evolution analysis) to make a statement that RdFV is more ancient than RGDV, which appears to have biased their question and approach (effect of RdFV on RGDV).

3. Based on the bias above in #2, the authors should consider the reciprocal analysis - the effect of RGDV on sperm infection, HONGrES1 binding, paternal transmission by RdFV. This reviewer suggests that the authors provide the reciprocal analysis underlying the 'cooperative' transmission analysis and conceptual model described in Figure 5 I-M. The outcome of the reciprocal silencing of P8 on CP transcripts and associated paternal transmission of RdFV may elucidate the cooperative or mutualistic nature of the virus-virus interaction.

4. The discussion adds very little new information beyond the results. It is more of a recapitulations of the text in the results. The authors should consider more synthesis of their findings - what is known about the phylogenetics/phylogenomics and host interactions with the Kitaviridae family members (Figure S1 shows placement of RdFV in this family). Are these plant viruses, insect viruses? The infection rate of the lab colony of leafhoppers was 80%? What does this mean about the 20%? Apparently they are just as fit as the infected insects, so are these facultative associates of the leafhopper or just residents? What are the escapes in the colony (-RdFV)? The authors did not provide TEM micrographs showing fertilized ova with RDGV or RdFV - however, are they aware of the efficiency of ova infection and what is known about male leafhopper insemination of females - the introduction the authors conjecture that males are more transmission efficient because they can mate more than once with more than one female. Can authors say anything new about this statement from their findings? Can the authors say anything about how their findings inform the transmission ecology of reovirus transmission in nature? OR the contribution of male vectors to transmission ecology of plant viruses? What new avenues of research open up for scientists with the new findings?

5. Statistical methods and assumptions were not stated for each type of variable - % transmission, relative transcript abundance, etc. State these methods and data transformations if used.

6. The composite manuscript figures are very densely packed and a few are misaligned and need adjustment. Are all subfigures necessary in the main manuscript? For example, Figure 1 is focused on reproductive organs, so the midgut TEM does not seem to add value for this particular Figure. Consider moving other tissue system analyses to supplemental data. Also, Figure 1 A-C could be it's own figure.

7. Order in Results: The authors start with the symbiont, but the introduction begins with the plant reovirus. Consider rearranging results or introduction based on the driving question - conserved mechanism of viral infection and transmission via sperm or effect of symbiont on plant virus transmission or cooperative transmission? Perhaps consider showing one Figure for both viruses with regards to TEM and infection, another Figure of protein-protein interactions with HONGrES1 for both viruses (singly), and another Figure for silencing (both viruses) etc. This is a suggestion. Since there are so many panels it's a challenging task.

8. Writing - Topic sentences in the results section are weak - instead of starting results paragraphs with methods, start with the logic or finding; several missing words (prepositions) and typos occurred throughout. There was inconsistent use of language or terms - example transmission electron microscopy and immunoelectron microscopy - both used when showing immunolocalization of virus and viral proteins. Be consistent. Figure titles should include the name of the leafhopper (genus species). Some legends do not, one says 'leafhoppers'. Ultrastructural components in micrographs are not clearly described in the legend. Are the spikes in the testes

sperm? The DAPI stain of sperm lights up the entire spike? Is the spikes the nuclei? Not all readers know the fine ultrastructure of reproductive organs of insects. Need more summary statistics in the results text for quantitative findings - Need info on % of reductions between treatments, or % of sperms showing the observed result (how representative is the one micrograph shown?), or what is meant by more frequent, or most or greater? To what extent? (See lines 179, 190, 201, 204, 208, 212 for examples)

9. The study is multi-layered and complex, and at times, the methods are challenging to follow - for example, throughout the methods, there is repeated use of the same matings - might the authors consider providing a table that shows the different matings (if they are different) and the response variables measured for each mating. Alternatively, rearrange methods text to have one description of matings (to reduce redundancy) and followed by the different measurements taken for each cross. Also, several details are missing in methods - primers used for detection and quantification, TEM settings, RT-qPCR reactions conditions, concentrations of components..

REVIEWER COMMENTS

Reviewer #2 (Remarks to the Author):

In this study, the authors identify a new sperm-specific serpin protein HongrES1 of leafhopper *Recilia dorsalis* for mediating paternal transmission of a rice reovirus and a novel symbiotic virus of the Kitaviridae family. This work is a continuation of the previous work by the same group where they that major outer capsid protein P8 of RGDV interacts with heparan sulfate proteoglycan (HSPG) to facilitate sperm-mediated paternal virus transmission. HongrES1 serves as the receptors to mediate the direct binding of virions to leafhopper sperm surfaces and subsequent paternal transmission via interaction with both viral capsid proteins. The study is well-conducted and the findings reveal a new mode for symbiotic viruses to synergistically hijack insect sperm-specific proteins for paternal transmission without disturbing sperm functions.

Overall, the manuscript is well-written and the figures are well-explained. Few comments below

Figure 1: It will be helpful to show the immunoelectron microscopy and Western blot analysis of the female RO as well for comparison.

Response: In the revised version, we have added Figure S3 of immunoelectron microscopy to shown RdFV infection in female RO, and Figure 2D of western blot assay to show the absence of HongrES1 accumulation in female RO. The related contents appeared on lines 110–111, 112–114, 147–149.

Figure S3 RdFV infection in ovaries of *R. dorsalis* leafhoppers.

Figure 2 (C, D) HongrES1 expression levels in different organs of 30 RdFV-free leafhoppers, as determined by RT-qPCR (C) and western blot (D) assay, respectively. Means (\pm SD) are shown from three biological replicates. *** P <0.001. WB, whole

bodies. ♂RO, male reproductive organs. ♀RO, female reproductive organs. AC, alimentary canals. SG, salivary glands. Res, residues.

Figure 2 and 3: Include quantitative data (Image J) from 3 or more independent expts for IFA images.

Response: In the revised version, we have carefully revised the Methods and Figure legends to show the quantitative data from 3 or more independent expts for IFA images. The related quantitative data of IFA images were shown in Figure 1H, 3C, 3F, 4B, 4D and 5L.

Figure 4: Including rescue experiments to support this data will strengthen the claim
 Response: The neutralizing virus-sperm binding by pretreatment with HongrES1 antibody or pre-immune antibody as the control. The knockdown of HongrES1 expression in male reproductive systems was performed by microinjection of synthesized dsHongrES1 or dsGFP (control) into newly emerged male adults. Currently, the rescue experiments are not available in virus-leafhopper interaction experiments.

Figure 5K: does dsCP affect the expression of HongrES1?

Response: In the revised version, we have added this experiment in Figure 6C.

Figure 6C Western blot assays showing the protein accumulation levels of RdFV CP, RGDV P8 and HongrES1 in 30 male reproductive organs of dsGFP-, dsCP or dsP8-treated RdFV and RGDV co-positive male adults. The proteins were detected by using HongrES1,

CP, P8 or histone H3 antibody, and relative intensities of bands are shown. Bands for histone H3 demonstrate loading of protein. Data represent three biological replicates.

IFA Co-localization indicates that majority of RGDV staining (red) doesn't co-localize with RdFV staining (green). To resolve this concern, a co-localization quantitative examination will be helpful.

Response: We have used the enlarged image to show clearly the colocalization of RGDV P8-formed puncta with RdFV CP-formed puncta or the distribution of P8-formed puncta on the edges of filamentous structures of CP. About half of P8-formed structures were colocalized with CP from infected 30 testes tested (n=30, 3 replicates). The related data are shown in Figure 5J-L and on lines 228-231.

Figure 5 (J-L) Colocalization of RdFV CP and RGDV P8 in male testes, as determined by immunofluorescence microscopy. The dissected RdFV and RGDV co-positive testes were immunostained with CP-FITC (green) and P8-rhodamine (red). ~50% of P8-formed structures were associated with CP in 30 testes (n=30, 3 replicates) (L). $**P < 0.01$. Panels K is the enlarged image of the boxed area in panels J. Te, testes. Ps, punctate structure. Fs, filamentous structures. Bars, 10 μ m.

Discussion: Maternal transmission may still be possible using another mechanism and hasn't been fully examined. This should be clarified. Authors explain that sperm function isn't affected by the two viruses. Has there been any morphological and functional analysis of the sperm?

Response: In the first paragraph of Discussion section, we have carefully discussed the possible mechanisms for maternal transmission via transovarial passage on lines 280-290.

In the revised version, we have added Figure S4 and Figure 3G of electron microscopy to show RdFV and RGDV did not affect sperm morphology. Furthermore, the experiments of effect of RdFV on *R. dorsalis* fitness in Figure S6 showed that RdFV infection benefited the fitness of host adults and their offspring, and thus did not affect the functions of host sperm or ovary. We have previously revealed the sperm-mediated paternal RGDV transmission without disturbing sperm functions (Mao et al., 2019). Thus, our electron microscopy and paternal transmission experiments confirmed that viral binding does not affect sperm morphological and functional characteristics. Related contents appeared on lines 115-116, 124-136, 182-183.

Figure S4 Effect of RdFV on sperm morphology as determined by electron microscopy. Testes of RdFV-free (A) or -positive (B) males were excised and processed for electron microscopy. Vi, virions. Bars, 100 nm.

Figure 3G Electron micrograph showing the sperms in virus-free testes. Bar, 500 nm.

Figure S6 Effects of vertical transmission of RdFV on fitness of *R. dorsalis* adults and offspring. Mating combinations were established as follows: infected virgin female × infected male, and uninfected virgin female × uninfected male. (A) Effects of RdFV infection on the longevity of male and female adults. Longevity of 50 RdFV-free or positive female or male adults was analyzed. Means (± SD) are shown from 50 insects, and represent three replicates. *P<0.05. (B–D) Progeny egg number (B), size (C), development duration (D) and hatching rate (E) of female adults from

different mating combinations.

Reference:

Mao, Q., Wu, W., Liao, Z., Li, J., Jia, D., Zhang, X., Chen, Q., Chen, H., Wei, J., Wei, T., 2019. Viral pathogens hitchhike with insect sperm for paternal transmission. *Nat Commun* 10, 955.

Reviewer #3 (Remarks to the Author):

This paper presents investigations of a largely unexplored mechanism for vertical arbovirus transmission, using viral binding to insect sperm receptors. The topic is very interesting and of significant importance as there is very limited knowledge on virus long term viral persistence, independent on availability of other viral hosts. I appreciate the investigations and comprehensive explanations, but the text lacks some important details of the work.

Introduction

The statement “scarcely affects insect fitness” is repeated twice (L41 and L47) but is only based on one reference, which is self-citation. The statement needs more support or less boldly expressed, and not repeated twice.

Response: We have deleted the sentence “However, arbovirus infection scarcely disturbs insect sperm functions.”

L49: Change to: “...arbovirus transmission may have evolved as a preferred mode of vertical transmission during the long-term virus-vector interaction.”

Response: We have changed this sentence as “Thus, paternal arbovirus transmission may have evolved as a preferred mode of vertical transmission during the long-term virus-vector interaction.”

L51: The sentence starts with “..insect symbiotic viruses” but in the following sentence you give ticks as an example. Use arthropod symbiotic viruses or remove ticks as an example as they are not insects.

Response: We have removed ticks.

L63: Do you mean “relatively complicated” (missed a d)? Specify, what do you mean with complicated components.

Response: We have deleted this sentence in the revised version.

Results

You have a lot of information in the figures that is lacking from the main text. Please add sample sizes, number of replicates and means, SDs, significance details etc to the main text (results/methods).

Response: We have carefully checked and revised the figures according to this comment. The related contents appeared in Methods, Figure legends and Results sections.

L91: How did you identify this virus and from where? I cannot find this as a reference, nor in any details in the methods.

Response: In the first paragraph of Methods section, we have carefully described the information of the identification of RdFV from field-caught *R. dorsalis* population in Guangdong Province, Southern China in 2018. The related contents appeared on lines 391-392.

L96: How many were tested?

Response: Our preliminary experiments using RT-PCR assay showed that about 80% of male (♂) or female (♀) *R. dorsalis* population (n=100, 3 replicates) reared under controlled greenhouse conditions had the transcript of RdFV CP (Figure 1B).

L101: by how many days?

Response: We have carefully described the experiments of effect of RdFV on *R. dorsalis* fitness in Figure S6, which showed that RdFV infection benefited the fitness of host adults and their offspring, and thus did not affect the functions of host sperm or ovary. The related contents appeared on lines 124-136, 512-530.

L135: plays

Response: We have changed this.

L139: To use the word titers is incorrect as you haven't titrated the virus and measured functional virions, you have only measured viral RNA with RT-PCR. It is an important distinction.

Response: We have revised this sentence as "RT-qPCR assay showed that the transcript levels of RGDV P8 were significantly higher in RdFV and RGDV co-positive male population than in RdFV-free and RGDV-positive male control (Figure 5A)." on lines 214-216.

Discussion

You have shown that the virus particles associate with the receptor but you don't discuss potential infection and replication. Could you elaborate on this discrepancy in the discussion.

Response: In the Discussion section, we carefully discussed the potential infection after the binding of virions to HongrES1 receptor on sperm surface. "In this case, the HongrES1 on sperm surfaces can be regarded as the receptor for RGDV or RdFV. However, such virus-sperm binding does not lead to the invading of virions into the sperm cytoplasm. Our electron microscopy and paternal transmission experiments confirm that viral binding does not affect sperm morphological and functional characteristics. Finally, infected males are able to venereally transmit RGDV or RdFV to females during mating, where the viruses are localized with the transferred sperms in the spermathecae. Such virus-decorated sperms in the female spermatheca finally move to the oviduct for fertilizing the eggs. Thus, HongrES1 serves as the

receptors to mediate the direct binding of virions to the sperm surfaces and subsequent paternal virus transmission (Figure 6G).” on lines 304–313.

L226–227 repeats L221–222 and needs editing.

Response: We have changed this.

L242–244 “We anticipate that other...” . Why do you anticipate this? Any references pointing to the same conclusion? Otherwise it is better to write “Potentially, other arboviruses...”

Response: We have changed this sentence as “Potentially, other arboviruses and symbiotic viruses might also have evolved to hitchhike with HongrES1 and its homologs on insect sperm surfaces for paternal transmission.” on lines 313–315.

Methods

Please provide more details on insect collection. When were insects and plants collected from the field? How was infection confirmed? Which greenhouse conditions?

Response: We have described this information in Methods section on lines 391–393, 423–425.

You use a lot of RT-PCR but there are no details of this PCRs. Please provide references and/or primers/cutoff values used.

Response: We have added Supplementary Table 1 to show these information. Related information was also shown in Methods section on lines 443–455.

L310–311: I don’ t understand this statement. Do you mean they were tested for the presence of RdFV to confirm infection? Or did you not know if they were infected when you set up the mating pairs?

Response: In the Methods section, we firstly established virus-positive or free leafhopper colony, which were used to perform vertical transmission experiments. After mating, the females and males were then tested by RT-PCR assay to further confirm uninfected or infected parents. The related contents appeared on lines 415–422 and lines 496–510.

L312–313: Here you write that offspring of crossing-pairs were raised to adults and then tested for RdFV. In the figure legends (L605–606, L667, L705) you write that eggs were collected. Which is correct?

Response: The eggs laid by 4 combinations were harvested at 7-day post oviposition by dissecting rice seedlings, and were individually placed on a piece of water-soaked filter paper in petri dishes at $25 \pm 3^\circ$ C. After eggs eclosion, offspring were individually fed on new rice seedlings, and the presence of RdFV or RGDV were also tested by RT-PCR assays. Thus, we changed to use “offspring” in the related experiments.

L326: give reference to method.

Response: We have revised this.

L355: Replace detected with tested: “the bodies were tested for”

Response: We have revised this.

L361; Replace detected with tested: “males were tested for”

Response: We have revised this.

Figures:

In general, the figures are too small, and with a too long legend text, making them unnecessarily difficult to follow. I recommend to divide them into several separate figures and condense the legend texts.

Response Thanks for this comment and we have followed.

L679, L699: “Approximate 30...”. How can numbers of tested in RT-qPCR be approximate?

Response: We have deleted “approximate” in related sentences.

S1: Why do you use neighbor-joining and not maximum likelihood or Bayesian methods? Support numbers not clearly visible and are also very low.

Response: We have used Bayesian methods to construct phylogenetic tree in Figure S1 in the revised vision. And reliable support numbers were also shown in the revised vision.

S2: What does the figure of the morphology of eggs say? To me they look the same. Legend of blue and red markings is incomplete.

Response: We have changed the figure of the morphology of eggs in Figure S6B. We also have used the new figures to show egg fitness after viral infection in Figure S6C-E.

Reviewer #4 (Remarks to the Author):

The study system presented by Wan et al. provides new pieces of molecular (functional proteomics, gene silencing) and ultrastructural cellular evidence to support one mechanism underlying the previously reported phenomenon of paternal transmission of a leafhopper-transmitted plant reovirus (rice gall dwarf virus, RGDV) via sperm to ova of female leafhoppers (rice green leafhopper, *Recilia dorsalis*). In addition, the authors discovered a new filamentous virus (RdFV) associated with the leafhopper in their lab colony (80% incidence). The authors conducted various mating experiments to test hypotheses to address the capability of this viral associate to be sexually-transmitted, if this viral associate affects transmission efficiency of the plant reovirus to leafhopper progeny and plants, and in parallel, performed several molecular and cellular analyses to determine if this new vector-associated virus (referred to 'symbiont' and 'endosymbiont' in the manuscript) utilizes the same mechanism of sperm attachment and transport to ova as the plant reovirus. The foundation for the study was well established previously by the authors and by others

examining paternal transmission by vector transmitted vertebrate viruses (mosquito) and other virologists studying other insect reoviruses and their nonstructural protein associated with sexual transmission (cited in references: Chen et al., 2019). Analogies with mammal sperm provided the focus on an insect ortholog of a serine protease inhibitor (serpin: HONGrES1) associated with sperm surface structure and function, which also formed the basis that this serpin may modulate (dampen) insect innate immunity: reduced phenol oxidase production and thus reduced melanization.

The methodologies used to generate evidence in support of the sperm-mediated mechanism (protein-protein interactions, transmission microscopy and immunolocalization of target proteins) of male-transmission are gold-standard methods in virology and microbe-host interactions, and the experimental design includes independent, replications of experiments and sufficient numbers of experimental units (insects) to provide robust analyses. While the findings regarding the shared mechanism of paternal transmission of the reovirus (RGDV) and symbiont (RdFV) are solid, there are some major concerns that need to be addressed explicitly to improve the quality of the study.

1. The authors conclude that the two viruses interact synergistically to 'hijack' sperm proteins for transmission. The use of the word 'synergy' and its derivatives in the title and results/discussion/figure legend text are not supported by the design of the experiment nor the analysis of the data. Several readers, including this reviewer, define synergy as: the combined effect of two entities (RGDV and RdFV) is significantly greater than the sum of their individual effects on a third entity (response of third entity). Since the authors are interested in how one virus (RdFV) affects the transmission of the other virus (RGDV), it is impossible to demonstrate synergy since one cannot show the single effect of RdFV (in absence of RGDV) on transmission of RGDV; RdFV must always be with RGDV to have an effect on RGDV transmission. Just for illustrative purposes, suppose sperm fitness was the response variable (3rd entity = sperm), then a synergic or additive or antagonistic effect could be demonstrated by: the combined effect of [RGDV infection (entity A) and RdFV infection (entity B)] on sperm fitness is greater/equal/less than sum of RGDV effect alone on fitness and RdFV effect alone on fitness, and there are statistical methods to distinguish additive from synergistic effects. In addition, the authors' interpretation of 'synergy' was based on results displayed in Figure 5 and perhaps Fig 3b(?): this reviewer does not interpret synergy based on the data in this figure. In figure 3b, one would reasonably interpret these data as RdFV enhancing transmission of RGDV, but one cannot conclude synergy between the viruses - statistical support for synergy is not possible without measuring a response of single entities, and in fact, it appears that RdFV increases rate of RGDV transmission from ~62% to ~79%. How can one say this is synergistic, additive or marginal? Please address this concern explicitly in the text - this reviewer does not define synergy between 2 entities as the effect of one entity on the other entity.

Response: We agree with this comment. We have deleted the “synergy”, “synergetic” and “synergistically”. In the revised version, we used “cooperatively” or “collaboratively” to show RdFV and RGDV hijack HongrES1 for simultaneous invasion into male reproductive organs and co-paternal transmission.

2. The authors presumed that RdFV (symbiont) facilitates or enhances sperm infection and paternal transmission by the plant reovirus (RGDV). Is the alternative possible: that RGDV facilitates, enhances sperm entry, and paternal transmission by the symbiont RdFV? The authors made a statement in the discussion (line 260) that ‘paternal transmission route that is established for the more ancient RdFV..’. The authors made no effort to determine the evolutionary history and age (molecular clock analysis) of RdFV vs RGDV. There is in fact, no discussion about this new virus and readers would be interested in knowing more about the membership (taxon) of this virus. The authors must provide evidence (molecular clock analysis or other virus evolution analysis) to make a statement that RdFV is more ancient than RGDV, which appears to have biased their question and approach (effect of RdFV on RGDV).

Response: It seems impossible for us to compare the Phylogenetic relationship between a dsRNA reovirus (RGDV) and an insect-specific filamentous (+) ssRNA virgavirus (RdFV). We thus deleted the related description that “RdFV facilitates paternal RGDV transmission via a direct synergistic interaction between both viral particles”.

Because RdFV or RGDV infection activates HongrES1 expression, and thus the knockdown of RGDV P8 or RdFV CP expression accordingly decreases HongrES1 accumulation, which finally inhibits viral infection in male reproductive system and subsequent paternal virus transmission (Figure 6A-F). Thus, our results suggest that the direct interaction of viral capsid proteins mediates simultaneous invasion of two viruses into male reproductive organs to attach sperms, and RdFV and RGDV could collaboratively activate HongrES1 for co-paternal transmission. The related contents appeared on lines 247-253.

Figure 6 Direct interaction between RdFV CP and RGDV P8 mediates their co-parental transmission by male *R. dorsalis*.

3. Based on the bias above in #2, the authors should consider the reciprocal analysis – the effect of RGDV on sperm infection, HONGrES1 binding, paternal transmission by RdFV. This reviewer suggests that the authors provide the reciprocal analysis underlying the 'cooperative' transmission analysis and conceptual model described in Figure 5 I–M. The outcome of the reciprocal silencing of P8 on CP transcripts and associated paternal transmission of RdFV may elucidate the cooperative or mutualistic nature of the virus–virus interaction.

Response: We agree with this comment and follow. We have added Figure 6A–F to show how the knockdown of RdFV CP expression inhibited RGDV P8 and HongrES1 accumulation in male reproductive organs and subsequent paternal RGDV transmission by co-infected insects. Similarly, the knockdown of RGDV P8 expression also inhibited RdFV CP and HongrES1 accumulation in male reproductive organs and subsequent paternal RdFV transmission by co-infected insects.

Because RdFV or RGDV infection activates HongrES1 expression, and thus the knockdown of RGDV P8 or RdFV CP expression accordingly decreases HongrES1 accumulation, which finally inhibits viral infection in male reproductive system and subsequent paternal virus transmission. Thus, our results suggest that the direct interaction of viral capsid proteins mediates simultaneous invasion of two viruses into male reproductive organs to attach sperms, and RdFV and RGDV could collaboratively activate HongrES1 for co-parental transmission. The related contents appeared on lines 240–253.

Figure 6 Direct interaction between RdFV CP and RGDV P8 mediates their co-parental transmission by male *R. dorsalis*.

4. The discussion adds very little new information beyond the results. It is more of a recapitulations of the text in the results. The authors should consider more

synthesis of their findings – what is known about the phylogenetics/phylogenomics and host interactions with the Kitaviridae family members (Figure S1 shows placement of RdFV in this family). Are these plant viruses, insect viruses? The infection rate of the lab colony of leafhoppers was 80%? What does this mean about the 20%? Apparently they are just as fit as the infected insects, so are these facultative associates of the leafhopper or just residents? What are the escapes in the colony (–RdFV)? The authors did not provide TEM micrographs showing fertilized ova with RDGV or RdFV – however, are they aware of the efficiency of ova infection and what is known about male leafhopper insemination of females. The introduction the authors conjecture that males are more transmission efficient because they can mate more than once with more than one female. Can authors say anything new about this statement from their findings? Can the authors say anything about how their findings inform the transmission ecology of reovirus transmission in nature? OR the contribution of male vectors to transmission ecology of plant viruses? What new avenues of research open up for scientists with the new findings?

Response: For the phylogenetics/phylogenomics and host interactions with the Virgaviridae family members. We have added the new paragraph in the Discuss section as “Phylogenetic analysis showed that the insect-specific RdFV, gramineous plants-infecting furoviruses, and insect-specific Hubei virga-like virus belong to Virgaviridae family (Adams et al., 2017; Kondo et al., 2019; Ramos-Gonzalez et al., 2022). The close phylogenetic relationship between insect-infecting virgaviruses and their plant-infecting counterparts suggests that they might share the common virus origin(s) (Kondo et al., 2019; Kondo et al., 2020). Plant virgaviruses are non-replicative in the insects (Adams et al., 2017; Creager, 2022; Olmedo-Velarde et al., 2021; Ramos-Gonzalez et al., 2022), while insect-specific RdFV and Hubei virga-like virus are also non-replicative in the plants (Kondo et al., 2019; Kondo et al., 2020), suggesting the existences of host-specific barriers for virgaviruses. In general, insect-specific symbiotic viruses such as Nilaparvata lugens reovirus and DcRV can be horizontally transmitted to their herbivorous insect hosts through plants (Chen et al., 2019; Kondo et al., 2019; Kondo et al., 2020; Nakashima and Noda, 1995). Although vertical transmission rate of RdFV by *R. dorsalis* population is not 100%, rice plants might also serve as the passive vectors for the horizontal transmission of RdFV, finally ensuring the effective spread of RdFV population in nature. Recent metagenomic studies have found that insect-specific virgaviruses are widespread in insects (Kondo et al., 2019), and may represent a major group of insect-specific paternally transmitted symbiotic viruses.” The related sentence appeared on lines 317-330.

For the paternal virus transmission, we have added Figure S5 of immunoelectron microscopy to show that virus-associated sperms were present in the dissected spermatheca of RdFV-free females at 5-day post mating with RdFV-positive males (Figure S5), confirming the transfer of virus-associated sperms from infected males to females. Mating experiments shown above (Figure 1F) suggested that such RdFV-associated sperms in the spermatheca finally fertilized the mature eggs during ovulation. We have previously revealed the presence of RGDV-associated sperms in the dissected spermatheca of RGDV-free females post mating with RGDV-positive males (Mao et al., 2019). The related sentence appeared on lines 116-121.

For the discussion on paternal virus transmission, we have carefully revised the related sentences related to the significance of our new findings. In particular, we added one new paragraph to show the transmission ecology of reovirus transmission in nature, as followed “Over the past 40 years, viral disease caused by RGDV is always epidemic in the field in Southern China. During the winter months (November to March) in Guangdong, Southern China (Jia et al., 2022), RGDV-positive *R. dorsalis* population can keep up to two generations on the weeds such as *Alopecurus aequalis* (Mao et al, 2019). A male has a huge number of sperms and can mate repeatedly with females to enhance virus spread. Furthermore, RGDV does not affect the fitness of male-transmitted offspring (Chen et al., 2016; Liao et al., 2017; Mao et al., 2019), but causes significant fitness cost of female-transmitted offspring (Mao et al., 2019). However, insect symbiotic RdFV is beneficial for adult host and offspring fitness, and thus facilitating vertical RGDV transmission. Thus, the high efficiency for paternal RGDV transmission through males and the low efficiency for transovarial RGDV transmission via males simultaneously ensure effective viral transmission and insect fecundity. We deduce that sperm-mediated paternal RGDV transmission is a powerful type of vertical virus transmission by insect vectors, and plays a vital role in the efficient maintenance of RGDV during the cold seasons in the field. Many arboviruses such as La Crosse virus and Zika virus can also be paternally transmitted by male mosquitoes (Thompson and Beaty, 1978; Li et al., 2017; Campos et al., 2017; Dahiya et al., 2022). Potentially, insect sperm-specific proteins or symbiotic viruses might also facilitate paternal arboviruses transmission ensure viral long-term persistence in nature.” The related sentence appeared on lines 359-374.

Figure S5 RdFV infection in spermatheca of female *R. dorsalis* at 5-day post mating with RdFV-positive males.

Figure 1F Vertical transmission rates of RdFV by RdFV-positive and free female or male leafhoppers via mating.

References:

- Adams, M. J., Adkins, S., Bragard, C., Gilmer, D., Li, D. W., MacFarlane, S. A., Wong, S. M., Melcher, U., Ratti, C., Ryu, K. H., Consortium, I. R., 2017. ICTV Virus Taxonomy Profile: Virgaviridae. J Gen Virol 98, 1999-2000.
- Campos, S. S., Fernandes, R. S., dos Santos, A. A. C., de Miranda, R. M., Telleria, E. L.,

- Ferreira-de-Brito, A., de Castro, M.G., Failloux, A.B., Bonaldo, M.C., Lourenco-de-Oliveira, R., 2017. Zika virus can be venereally transmitted between *Aedes aegypti* mosquitoes. *Parasite Vector* 10.
- Chen, Q., Godfrey, K., Liu, J., Mao, Q., Kuo, Y.W., Falk, B.W., 2019. A Nonstructural Protein Responsible for Viral Spread of a Novel Insect Reovirus Provides a Safe Channel for Biparental Virus Transmission to Progeny. *J Virol* 93.
- Creager, A.N.H., 2022. Tobacco Mosaic Virus and the History of Molecular Biology. *Annu Rev Virol* 9, 39–55
- Dahiya, N., Yadav, M., Yadav, A., Sehrawat, N., 2022. Zika virus vertical transmission in mosquitoes: A less understood mechanism. *J Vector Borne Dis* 59, 37–44.
- Jia, D., Luo, G., Shi, W., Liu, Y., Liu, H., Zhang, X., Wei, T., 2022. Rice Gall Dwarf Virus Promotes the Propagation and Transmission of Rice Stripe Mosaic Virus by Co-infected Insect Vectors. *Front Microbiol* 13, 834712.
- Kondo, H., Chiba, S., Maruyama, K., Andika, I.B., Suzuki, N., 2019. A novel insect-infecting virga/nege-like virus group and its pervasive endogenization into insect genomes. *Virus Research* 262, 37–47.
- Kondo, H., Fujita, M., Hisano, H., Hyodo, K., Andika, I.B., Suzuki, N., 2020. Virome Analysis of Aphid Populations That Infest the Barley Field: The Discovery of Two Novel Groups of Nege/Kita-Like Viruses and Other Novel RNA Viruses. *Frontiers in Microbiology* 11.
- Li, C.X., Guo, X.X., Deng, Y.Q., Xing, D., Sun, A.J., Liu, Q.M., Wu, Q., Dong, Y.D., Zhang, Y.M., Zhang, H.D., Cao, W.C., Qin, C.F., Zhao, T.Y., 2017. Vector competence and transovarial transmission of two *Aedes aegypti* strains to Zika virus. *Emerg Microbes Infect* 6.
- Liao, Z., Mao, Q., Li, J., Lu, C., Wu, W., Chen, H., Chen, Q., Jia, D., Wei, T., 2017. Virus-induced tubules: a vehicle for spread of virions into ovary oocyte cells of an insect vector. *Front Microbiol* 8, 475.
- Mao, Q., Wu, W., Liao, Z., Li, J., Jia, D., Zhang, X., Chen, Q., Chen, H., Wei, J., Wei, T., 2019. Viral pathogens hitchhike with insect sperm for paternal transmission. *Nat Commun* 10, 955.
- Nakashima, N., Noda, H., 1995. Nonpathogenic *Nilaparvata lugens* reovirus is transmitted to the brown planthopper through rice plant. *Virology* 207, 303–307.
- Olmedo-Velarde, A., Hu, J., Melzer, M.J., 2021. A Virus Infecting *Hibiscus rosa-sinensis* Represents an Evolutionary Link Between Cileviruses and Higreviruses. *Frontiers in Microbiology* 12.
- Ramos-Gonzalez, P.L., Chabi-Jesus, C., Tassi, A.D., Calegario, R.F., Harakava, R., Nome, C.F., Kitajima, E.W., Freitas-Astua, J., 2022. A Novel Lineage of Cile-Like Viruses Discloses the Phylogenetic Continuum Across the Family Kitaviridae. *Frontiers in Microbiology* 13.
- Thompson, W.H., Beaty, B.J., 1978. Venereal transmission of La Crosse virus from male to female *Aedes triseriatus*. *Am J Trop Med Hyg* 27, 187–196.

5. Statistical methods and assumptions were not stated for each type of variable – % transmission, relative transcript abundance, etc. State these methods and data

transformations if used.

Response: We have carefully checked and revised the full text according to this comment. The related contents appeared in Methods, Figure legends and Results sections.

6. The composite manuscript figures are very densely packed and a few are misaligned and need adjustment. Are all subfigures necessary in the main manuscript? For example, Figure 1 is focused on reproductive organs, so the midgut TEM does not seem to add value for this particular Figure. Consider moving other tissue system analyses to supplemental data. Also, Figure 1 A-C could be it's own figure.

Response: We agree with this comment and followed. We have carefully modified all figures including Figure 1.

7. Order in Results: The authors start with the symbiont, but the introduction begins with the plant reovirus. Consider rearranging results or introduction based on the driving question – conserved mechanism of viral infection and transmission via sperm or effect of symbiont on plant virus transmission or cooperative transmission? Perhaps consider showing one Figure for both viruses with regards to TEM and infection, another Figure of protein-protein interactions with HONGRES1 for both viruses (singly), and another Figure for silencing (both viruses) etc. This is a suggestion. Since there are so many panels it's a challenging task.

Response: We agree with this comment and followed. We have carefully checked and revised the full text, including the beginning with symbiotic viruses and then arboviruses in Introduction, and the combination of two viruses in one story in Results.

8. Writing – Topic sentences in the results section are weak – instead of starting results paragraphs with methods, start with the logic or finding; several missing words (prepositions) and typos occurred throughout. There was inconsistent use of language or terms – example transmission electron microscopy and immunoelectron microscopy – both used when showing immunolocalization of virus and viral proteins. Be consistent. Figure titles should include the name of the leafhopper (genus species). Some legends do not, one says 'leafhoppers'. Ultrastructural components in micrographs are not clearly described in the legend. Are the spikes in the testes sperm? The DAPI stain of sperm lights up the entire spike? Is the spikes the nuclei? Not all readers know the fine ultrastructure of reproductive organs of insects. Need more summary statistics in the results text for quantitative findings – Need info on % of reductions between treatments, or % of sperms showing the observed result (how representative is the one micrograph shown?), or what is meant by more frequent, or most or greater? To what extent? (See lines 179, 190, 201, 204, 208, 212 for examples).

Response: We have carefully revised the Results section to show clearly the methods, logic and finding in one paragraph.

For electron microscopy and immunoelectron microscopy, we have described these two different methods in Methods section in detail. Electron microscopy is used for observation of viral particles, while immunoelectron microscopy is used for localization of viral or insect proteins using antibodies and goat anti-rabbit IgG conjugated with 15-nm diameter gold particles as the second antibody. The related contents appeared on lines xx-xx.

We also have carefully revised the figure legends according to the comments, including figure titles and the ultrastructural components in micrographs.

For statistics in the results text for quantitative findings, we also have carefully revised the full text according to this comment, including Methods, Figure legends, Figures and Results sections.

9. The study is multi-layered and complex, and at times, the methods are challenging to follow - for example, throughout the methods, there is repeated use of the same matings - might the authors consider providing a table that shows the different matings (if they are different) and the response variables measured for each mating. Alternatively, rearrange methods text to have one description of matings (to reduce redundancy) and followed by the different measurements taken for each cross. Also, several details are missing in methods - primers used for detection and quantification, TEM settings, RT-qPCR reactions conditions, concentrations of components..

Response: We have carefully revised the Methods section according to this comment. In the Methods section, we firstly established virus-positive or free leafhopper colony, which were used to perform vertical transmission experiments. After mating, the females and males were then tested by RT-PCR assays to further confirm uninfected or infected parents. The related contents appeared on lines xx-xx and lines 415-422, 496-510.

For the missing information in methods, we also have carefully revised, including adding three Supplementary Tables to show the primers used in PCR and qPCR, and the mating combination used. Furthermore, the TEM settings and concentrations of components also have been carefully added.

REVIEWERS' COMMENTS

Reviewer #2 (Remarks to the Author):

The authors have addressed most of the comments raised by this and other reviewers. New quantitative data and changes in the discussion have strengthened this interesting study.

Reviewer #4 (Remarks to the Author):

The revised manuscript is much improved and the authors were mindful with addressing major concerns presented by the referees' comments. I am pleased with modifications made to enhance the quality of this work and manuscript. This work will be a major contribution to the understanding of the dynamic and unique features underscoring vector transmission biology and molecular mechanisms of plant virus-insect vector interactions.

REVIEWERS' COMMENTS

Reviewer #2 (Remarks to the Author):

The authors have addressed most of the comments raised by this and other reviewers. New quantitative data and changes in the discussion have strengthened this interesting study.

Response: Thank you.

Reviewer #4 (Remarks to the Author):

The revised manuscript is much improved and the authors were mindful with addressing major concerns presented by the referees' comments. I am pleased with modifications made to enhance the quality of this work and manuscript. This work will be a major contribution to the understanding of the dynamic and unique features underscoring vector transmission biology and molecular mechanisms of plant virus-insect vector interactions.

Response: Thank you.